# Fairness Guarantees under Demographic Shift

**Stephen Giguere**
Department of Computer Science, University of Texas at Austin
`sgiguere@cs.utexas.edu`

**Blossom Metevier, Yuriy Brun, Bruno Castro da Silva, & Philip S. Thomas**
College of Information and Computer Sciences, University of Massachusetts

**Scott Niekum**
Department of Computer Science, University of Texas at Austin

## Abstract

Recent studies found that using machine learning for social applications can lead to injustice in the form of racist, sexist, and otherwise unfair and discriminatory outcomes. To address this challenge, recent machine learning algorithms have been designed to limit the likelihood such unfair behavior occurs. However, these approaches typically assume the data used for training is representative of what will be encountered in deployment, which is often untrue. In particular, if certain subgroups of the population become more or less probable in deployment (a phenomenon we call *demographic shift*), prior work's fairness assurances are often invalid. In this paper, we consider the impact of demographic shift and present a class of algorithms, called `Shifty` algorithms, that provide high-confidence behavioral guarantees that hold under demographic shift when data from the deployment environment is unavailable during training. `Shifty`, the first technique of its kind, demonstrates an effective strategy for designing algorithms to overcome demographic shift's challenges. We evaluate `Shifty` using the UCI Adult Census dataset (Kohavi and Becker, 1996), as well as a real-world dataset of university entrance exams and subsequent student success. We show that the learned models avoid bias under demographic shift, unlike existing methods. Our experiments demonstrate that our algorithm's high-confidence fairness guarantees are valid in practice and that our algorithm is an effective tool for training models that are fair when demographic shift occurs.

## 1 Introduction

As machine learning (ML) algorithms continue to be used to aid decisions in socially impactful applications (Angwin et al., 2016; Goodall, 2016; Olson, 2011), it is becoming increasingly important to ensure that trained models are able to avoid bias and discrimination. Observations that the use of ML algorithms might have unexpected social implications, such as racial bias, have led to the creation of algorithms that provide high-confidence fairness guarantees (Thomas et al., 2019; Agarwal et al., 2018). These guarantees rely on the assumption that the data the model is trained with and the data encountered after deployment follow the same distribution. However, this assumption is false for many real-world problems (Zhuang et al., 2021), and, as we demonstrate, models often violate fairness guarantees and exhibit unfair bias when evaluated on data from a different distribution.

As an example, consider a model that uses university entrance exam scores to predict subsequent success. Because student demographics, such as race or sex, can shift over time, the distribution of applicants may change between model training and deployment. As a result, even if the model is trained to protect a disadvantaged group, if the learning algorithm assumes the training and deployment distributions are the same, many fairness guarantees will not hold (Section 4 empirically verifies this). We refer to such distribution change as *demographic shift*. Specifically, demographic shift occurs when the difference between the training and deployment distributions can be explained

by a shift in the marginal distribution of a single random variable, such as race or sex. Furthermore, demographic shift can often occur after a predictive model has been trained, so it is often the case that such shift must be accounted for without access to data from the deployment distribution.

We present `Shifty`, the first strategy for designing ML algorithms that provide high-confidence guarantees that one or more user-specified fairness constraints will hold despite a demographic shift between training and deployment without using data from the deployment environment. We design `Shifty` algorithms to work in two scenarios based on what the user knows at training time about the demographic shift: **(1)** the old and new demographic proportions are known, or **(2)** the demographic proportions are bounded in known intervals.

We evaluate `Shifty` on the UCI Adult Census dataset (Kohavi and Becker, 1996), as well as a real-world dataset of students' college entrance exam scores and their subsequent grade point average (GPA) in the Brazilian university system (da Silva, 2019). We compare `Shifty` to three families of existing algorithms: Fairness Constraints (Zafar et al., 2017), does not provide fairness without guarantees, Seldonian algorithms (Thomas et al., 2019) and Fairlearn (Agarwal et al., 2018), both of which provide high-confidence fairness guarantees, and RFLearn (Du and Wu, 2021).

`Shifty` allows the user to provide (one or multiple simultaneous) fairness definitions from a large class appropriate to the application domain, such as demographic parity, disparate impact, equalized odds, predictive equality, and individual fairness. Unlike all prior algorithms, `Shifty` algorithms provide high-confidence guarantees that the learned model satisfies these fairness constraints even when deployed on a distribution different from the training one. We demonstrate empirically that models learned by previous algorithms do, at times, violate the desired properties under demographic shift, while `Shifty`'s models do not. If there is insufficient training data, or if the specified fairness properties are simultaneously unsatisfiable, `Shifty` algorithms are designed to return no solution, but we show that this rarely happens in practice. Finally and crucially, we demonstrate empirically that, in our evaluated domain, `Shifty` is able to learn models that exhibit no loss of accuracy compared to the models that do not guarantee fairness, as long as sufficient training data exists.

Our main contributions are: **(1)** the first classification algorithms that provide high-confidence fairness guarantees under demographic shift, **(2)** a constructive proof of the guarantees, and a method for creating such algorithms, **(3)** an evaluation on real-world data, and **(4)** an open-source `Shifty` implementation and a release of all our data.

## 2  BACKGROUND AND RELATED WORK

We illustrate our approach for fair classification, although the methods we propose are easily extended to other problem settings (Thomas et al., 2019; Metevier et al., 2019). In this setting, a data *instance* consists of a set of *features* and an associated *label*. When considering the fairness of a classifier, each instance can be augmented with a *fairness attribute*. This information is often not used for prediction (e.g., some laws prohibit the use of race or sex in hiring decisions), but is assumed to be available for determining if the classifier exhibits bias.

We denote features by $X \in \mathcal{X}$, labels by $Y \in \mathcal{Y}$, and the fairness attribute by $S \in \mathcal{S}$. We assume that $(X, Y, S)$ is sampled from some joint probability distribution defined over $\mathcal{X} \times \mathcal{Y} \times \mathcal{S}$. While $\mathcal{S}$ generally represents a categorical protected attribute, such as race or sex, our formulation does not require that $\mathcal{S}$ be discrete. The naïve classification setting ignores the fairness attribute, and the goal is to accurately predict the label associated with $X$ when its true label is unknown. These predictions are generated using a *model*, $\theta : \mathcal{X} \to \mathcal{Y}$. A loss function, such as expected classification error, measures the quality of $\theta$. To obtain an accurate classifier, one typically selects a training algorithm, $a$, designed to minimize the chosen loss, and supplies it with a dataset consisting of $n$ observations sampled independently from the joint distribution—that is, $D = \{(X_i, Y_i, S_i)\}_{i=1}^n$, where $\Pr(X_i, Y_i, S_i) := \Pr(X, Y, S)$ for all $i \in \{1, ..., n\}$.

To assess the fairness of an algorithm, the user specifies a function, $g$, which accepts a model and is calibrated so that $g(\theta) > 0$ if and only if $\theta$ behaves unfairly. Typically, $g$ depends on the fairness attribute, $S$. For example, to assess if a classifier is biased based on race, $g$ might measure the difference in the classifier's accuracy for individuals of one race compared to another.

Importantly, $g$ can be defined based on the particular fairness requirements of a given application. Here, we consider the illustrative case where $g(\theta)$ is based on conditional expected value. (Appendix C.2 extends our approach to handle more general definitions of $g$.) Concretely, let $H := h(X, Y, S, \theta)$ define some choice of real-valued observable, let $\xi := c(X, Y, S, \theta)$ be some event, and let $\tau$ represent a real-valued tolerance. We then assume that $g$ is defined by $g(\theta) := \mathbf{E}[H \mid \xi] - \tau$. For example, if for a binary classification problem, the fairness attribute is sex, and the user wants to ensure that the false-positive rate of the model is below $20\%$ for females, one might set $g(\theta) = \mathbf{E}[\theta(X) \mid Y=0, S=\texttt{female}] - 0.2$. While this form of $g$ is not flexible enough to represent many widely used notions of fairness, it illustrate our strategies for accounting for demographic shift, which can easily be applied to more sophisticated definitions (see Appendix C.2).

Given a definition for $g$, we say that a training algorithm, $a$, is fair with high confidence if

$$\Pr\big(g(a(D)) \leq 0\big) \geq 1-\delta, \tag{1}$$

for some confidence threshold, $\delta \in [0, 1]$. Allowing the user to set $\delta$ during training overcomes the problem that guaranteeing fairness with absolute certainty is often impossible (Thomas et al., 2019).

It is possible that for some $g$ provided by the user, there is no model $\theta$ that satisfies $g(\theta) \leq 0$ and, consequently, no algorithm $a$ that satisfies (1). To address this, we adopt the convention described by Thomas et al. (2019) and permit algorithms to return NO_SOLUTION_FOUND (NSF), which is assumed to be fair by definition—that is, we assume that $g(\texttt{NSF}) = 0$. Intuitively, if a fair predictive model cannot be found, such algorithms are permitted to abstain from outputting an unfair predictive model, instead alerting the user that the fairness constraints could not be enforced with the required probability using the data provided. Since the trivial algorithm, $a(D) = \texttt{NSF}$ satisfies (1) by definition, we seek algorithms that satisfy (1) but return useful predictive models as frequently as possible, and we explicitly evaluate this consideration in our experimental designs.

**Fair Classification under Demographic Shift:** To reason about differences between the training and deployment data distributions, we augment each data instance with a random variable representing a *demographic attribute*, denoted by $T \in \mathcal{T}$. The demographic attribute is often distinct from the other variables defining each observation, but does not need to be. For example, a user might seek a model that avoids unfair bias against individuals of a particular sex, even if the distribution of the individuals' race differs from the training distribution. In this case, $S$ would represent the sex of an individual, while $T$ would represent their race.

Given the demographic attribute, we let $(X, Y, S, T)$ represent an instance observed during training and let $(X', Y', S', T')$ represent an instance encountered once the model is deployed. To formalize the effect of demographic shift, we assume that the demographic attribute's marginal distribution may change between training and deployment, but that the pre- and post-shift joint distributions over the instances are otherwise identical. This can be summarized by the following two conditions, which we refer to as the *demographic shift assumptions*:

$$\exists\, t \in \mathcal{T} \quad \text{s.t.} \quad \Pr(T = t) \neq \Pr(T' = t), \quad \text{and} \tag{2}$$

$$\forall\,(x, y, s, t), \quad \Pr(X{=}x, Y{=}y, S{=}s \mid T{=}t) = \Pr(X'{=}x, Y'{=}y, S'{=}s \mid T'{=}t). \tag{3}$$

Because the fairness of a model—that is, the value of $g(\theta)$—implicitly depends on $X$, $Y$, and $S$, it follows that guarantees of fairness based on $g$ may fail to hold after the model is deployed, which corresponds to replacing these random variables with $X'$, $Y'$ and $S'$. Formally, if $H' = h(X', Y', S', \theta)$ and $\xi' = c(X', Y', S', \theta)$, so that $g'(\theta) = \mathbf{E}[H' \mid \xi'] - \tau$ measures the prevalence of unfair behavior after $\theta$ is deployed, then the challenge presented by demographic shift is summarized by the observation that, for all training algorithms $a$,

$$\underbrace{\Pr\big(g(a(D)) \leq 0\big) \geq 1-\delta}_{\text{Property A}} \quad \not\Longrightarrow \quad \underbrace{\Pr\big(g'(a(D)) \leq 0\big) \geq 1-\delta}_{\text{Property B}}.$$

Therefore, we address the following problem:

**Problem Statement:** Given a user's description of possible demographic shift that might be present between the training and deployment environments as well as one or more definitions of fairness, design an algorithm, $a$, that provides high-confidence fairness guarantees that a returned model will behave fairly once the model is deployed—that is, an algorithm that satisfies Property B.

In this paper, we assume the user's description of the demographic shift is defined by a set of upper and lower bounds on the marginal probability of each value of the demographic attribute after deployment. Specifically, the user provides an input, $\mathcal{Q} := \{(a_t, b_t)\}_{t \in \mathcal{T}}$, encoding the assumption that $\Pr(T'{=}t) \in [a_t, b_t]$ for all $t \in \mathcal{T}$. However, the approach used by `Shifty` is general and can be applied for other descriptions of demographic shift as well. Given $\mathcal{Q}$ of this form, we identify two settings in which `Shifty` can be applied. In the case of *known demographic shift*, which occurs when the user knows the exact post-deployment distribution over the demographic attribute, the user sets $a_t = b_t = \Pr(T'{=}t)$ for each $t \in \mathcal{T}$, and `Shifty` will be guaranteed not to output unfair models under the specified deployment distribution. In the second setting, the user does not know the exact demographic shift that will occur and instead specifies non-singleton intervals for each $\Pr(T'{=}t)$. We refer to this as the *unknown demographic shift* setting.

**Related Work:** While many strategies promote fairness in ML models, most do not offer fairness guarantees, and none provide guarantees under demographic shift. Appendix A discusses existing approaches in detail. These approaches include methods based on enforcing hard constraints (Irani, 2015), soft constraints (Zafar et al., 2017; Smits and Kotanchek, 2005), chance-constrained programming techniques (Charnes and Cooper, 1959; Miller and Wagner, 1965; Prékopa, 1970), or by pre- or post-processing training dataset and model predictions (Verma et al., 2021; Salimi et al., 2019; Awasthi et al., 2020). The closest approaches to ours are Seldonian algorithms (Thomas et al., 2019), which allow the user to specify the fairness definition and provide high-confidence guarantees—that is, they satisfy Property A. Various approaches have been proposed to promote fairness in under particular types of distribution that differ from demographic shift (Lipton et al., 2018), such as *concept shift* (Schumann et al., 2019), *label shift* and *label bias* (Dai, 2020), and *prior probability shift* (Biswas and Mukherjee, 2021). The most similar problem setting to demographic shift is *covariate shift*, which occurs when the distribution of fairness attributes and feature vectors changes between training and deployment, but the conditional distribution over labels is unchanged. In addition to the differences between covariate shift and demographic shift (see Appendix A), the approaches that overcome covariate shift are not proven to satisfy Property B and, unlike `Shifty` algorithms, require access to either data from the deployment distribution or a complete causal model of the problem setting (Rezaei et al., 2021; Coston et al., 2019; Singh et al., 2021). One exception to this is RFLearn, which promotes fair outcomes under covariate shift without access to data from the deployment environment, but does not provide fairness guarantees (Du and Wu, 2021).

## 3 METHODOLOGY

An overview of `Shifty` is shown in Figure 1. Motivated by the design principles of previous Seldonian algorithms (Thomas et al., 2019), which are effective at designing algorithms that satisfy Property A, `Shifty` algorithms consist of three core components: *data partitioning*, *candidate selection*, and a *fairness test*. First, the data partitioning step splits the input dataset into two parts, which are used to perform the candidate selection and fairness test steps, respectively (see Appendix C.1). Next, a candidate model is trained. In practice, this can be performed using any existing classification algorithm, as the candidate selection step is not responsible for establishing the fairness guarantees `Shifty` provides. Once a candidate model is found, `Shifty` performs a fairness test by computing a high-confidence upper bound on the prevalence of unfair behavior when the candidate model is deployed in an environment affected by the demographic shift described by $\mathcal{Q}$. If this high-confidence upper bound is below zero, the candidate model is likely to behave fairly once deployed, and the candidate is re-

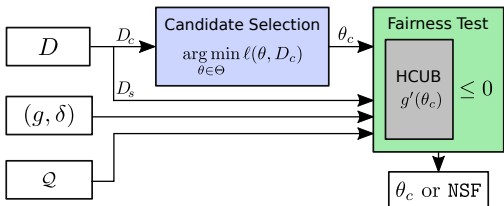

Figure 1: `Shifty` accepts training data, $D$, one or more fairness specifications (each consisting of a definition of unfair behavior, $g$, and a tolerance, $\delta$), and a description of possible demographic shifts, $\mathcal{Q}$. It first partitions $D$ into $D_c$ and $D_f$, and uses $D_c$ to select a candidate model, $\theta_c$. Then, it uses $\mathcal{Q}$ and $D_f$ to compute a $(1{-}\delta)$-probability high-confidence upper bound (HCUB) on the value of $g(\theta_c)$ after deployment, for each fairness definition. If these upper bounds are below zero, `Shifty` returns $\theta_c$, and otherwise returns `NSF`.

---

**Algorithm 1** `Shifty`$(D, g, \delta, \mathcal{Q})$

---

1: $D_c, D_f \leftarrow$ `Partition`$(D)$
2: $\theta_c \leftarrow$ `TrainCandidate`$(D_c, g, \delta, \mathcal{Q})$
3: $u \leftarrow$ `HighConfUB`$(\theta_c, g, D_f, \delta, \mathcal{Q})$
4: **return** $\theta_c$ **if** $u \leq 0$ **else** `NSF`

---

turned. However, if the value of the high-confidence upper bound is greater than zero, then `Shifty` cannot show that the candidate model will behave fairly with the required confidence, and returns `NSF` instead. Because `Shifty` only returns candidate models if they can be shown to be fair with high confidence on the demographic-shifted deployment distribution, it is guaranteed to satisfy Property B.

Algorithm 1 presents high-level pseudocode for classification algorithms that provide high-confidence fairness guarantees under demographic shift. `TrainCandidate` implements the candidate selection step, and `HighConfUB` implements the high-confidence upper bound on $g'(\theta_c)$ that is used to determine the output of the algorithm and to establish its theoretical guarantees. In the following sections, we describe the candidate selection and fairness test steps in detail. Because it is the fairness test that causes `Shifty` to satisfy Property B, we outline this component first.

## 3.1 THE SHIFTY FAIRNESS TEST

Given a candidate model, $\theta_c$, `Shifty`'s fairness test computes a $(1-\delta)$-confidence upper bound on $g'(\theta_c)$, which measures the prevalence of unfair behavior once $\theta_c$ is deployed. Therefore, the primary challenge associated with designing an algorithm that satisfies Property B is computing a valid high-confidence upper bound on $g'(\theta_c)$ given $\mathcal{Q}$, the description of the possible demographic distributions that might be encountered upon deployment.

Here, we propose a strategy for computing high-confidence upper bounds on $g'(\theta_c)$, starting with the simpler case in which the exact demographic shift is known and then extending this approach to the case in which the shift is unknown. In both settings, we show how to compute these bounds using appropriate confidence intervals. Our strategy is general and can be applied starting with many different confidence intervals. For illustrative purposes, we derive bounds based on inversion of the Student's $t$-test (Student, 1908), leading to an implementation of `Shifty` which we call `Shifty-ttest`. Importantly, our use of the Student's $t$-test implies that the resulting high-confidence upper bounds only hold exactly if $\Pr(H'|\xi)$ is a normal distribution.

### 3.1.1 KNOWN DEMOGRAPHIC SHIFT

To begin, we consider the task of assessing unfair behavior when there is no demographic shift. Given $n$ i.i.d. samples $\{(H_i, \xi_i)\}_{i=1}^n$, where $H_i \coloneqq h(X_i, Y_i, S_i, \theta)$ and $\xi_i \coloneqq c(X_i, Y_i, S_i, \theta)$, one can derive $U_{\texttt{ttest}}$, a function that computes a high-confidence upper bound on $\mathbf{E}[H|\xi]$, by inverting the commonly-used Student's $t$-Test (Student, 1908):

$$\Pr\left(\mathbf{E}[H|\xi] \leq U_{\texttt{ttest}}(g, D, \theta, \delta)\right) \geq 1-\delta. \tag{4}$$

Specifically, if $\mathcal{I}_\xi$ are the indices of the samples for which $\xi_i = \texttt{True}$ and $N_\xi = |\mathcal{I}_\xi|$, then

$$U_{\texttt{ttest}}(g, D, \theta, \delta) \coloneqq \frac{1}{N_\xi} \sum_{i \in \mathcal{I}_\xi} H_i + \frac{\sigma(g, D, \theta)}{\sqrt{N_\xi}} t_{1-\delta, N_\xi - 1},$$

where $t_{1-\delta, N_\xi - 1}$ is the $1-\delta$ quantile of the Student's $t$ distribution with $N_\xi - 1$ degrees of freedom, and $\sigma$ computes the sample standard deviation using Bessel's correction. As $g(\theta) = \mathbf{E}[H|\xi] - \tau$, it follows that in the absence of demographic shift, Property A is satisfied if algorithm $a$ only returns models satisfying $U_{\texttt{ttest}}(g, D, \theta, \delta) - \tau \leq 0$, and otherwise returns `NSF`.

To provide fairness guarantees that hold under demographic shift, we require a strategy for testing whether or not $g'(\theta) \leq 0$ with high probability, where $g'(\theta) = \mathbf{E}[H'|\xi'] - \tau$. However, $H'$ and $\xi'$ are defined with respect to the demographic-shifted distribution, for which no samples are available. Thus, we seek a new random variable, $\hat{H}$, that can be computed from $X$, $Y$, and $S$, but which satisfies $\mathbf{E}[\hat{H}|\xi] = \mathbf{E}[H'|\xi']$. Under the demographic shift assumptions, the reweighted variable defined by $\hat{H} \coloneqq \phi(T)H$ satisfies these requirements, where $\phi$ is an *importance weight* derived in Appendix B:

$$\phi(t) \coloneqq \frac{\Pr(T'{=}t|\xi')}{\Pr(T{=}t|\xi)} = \frac{\Pr(\xi|T{=}t)\Pr(T'{=}t)}{\Pr(T{=}t|\xi)\sum_{t'\in\mathcal{T}}\Pr(\xi|T{=}t')\Pr(T'{=}t')}, \tag{5}$$

for all $t \in \mathcal{T}$. Specifically, $\phi(T)$ acts as a scaling factor that reweights samples obtained during training so that their sample mean is an unbiased estimator of $\mathbf{E}[H'|\xi']$, as shown by Theorem 1.

**Theorem 1.** *Assume that $\Pr(T{=}t) \geq 0$ for all $t \in \mathcal{T}$. If the demographic shift properties hold, then the random variable $\hat{H} := \phi(T)H$ satisfies $\mathbf{E}[\hat{H}|\xi] = \mathbf{E}[H'|\xi']$, where $\phi$ is defined by (5).* **Proof.** *See Appendix B.*

Because $\hat{H}$ is defined with respect to pre-shift random variables, it is possible to generate i.i.d. samples of $\hat{H}$ during training, even when no samples from the deployment distribution are available. In particular, a set of i.i.d. observations of $\hat{H}$ is obtained by computing $\{\hat{H}_i\}_{i \in \mathcal{I}_\xi}$, where each $\hat{H}_i = \phi(T_i)h(X_i, Y_i, S_i, \theta)$. Using $\{\hat{H}_i\}_{i \in \mathcal{I}_\xi}$, we apply the inversion of the Student's $t$-test to derive $\hat{U}_{\texttt{ttest}}(g, D, \theta, \delta)$, which satisfies $\Pr(\mathbf{E}[\hat{H}|\xi] \leq \hat{U}_{\texttt{ttest}}(g, D, \theta, \delta)) \geq 1{-}\delta$. Specifically, if $\hat{\sigma}$ denotes the sample standard deviation of the reweighted observations, $\{\hat{H}_i\}_{i \in \mathcal{I}_\xi}$, then

$$\hat{U}_{\texttt{ttest}}(g, D, \theta, \delta) := \frac{1}{N_\xi} \sum_{i \in \mathcal{I}_\xi} \hat{H}_i + \frac{\hat{\sigma}(g, D, \theta)}{\sqrt{N_\xi}} t_{1-\delta, N_\xi - 1}. \tag{6}$$

Since $\mathbf{E}[\hat{H}|\xi] = \mathbf{E}[H'|\xi']$ by Theorem 1, it follows that $\hat{U}_{\texttt{ttest}}$ is also a high-confidence upper bound suitable for assessing fairness after demographic shift:

$$\Pr\left(\mathbf{E}[\hat{H}|\xi] \leq \hat{U}_{\texttt{ttest}}(g, D, \theta, \delta)\right) = \Pr\left(\mathbf{E}[H'|\xi'] \leq \hat{U}_{\texttt{ttest}}(g, D, \theta, \delta)\right) \geq 1{-}\delta. \tag{7}$$

Recalling that $g'(\theta) := \mathbf{E}[H'|\xi'] - \tau$, it follows that $g'(\theta) \leq 0$ with high confidence if $\hat{U}_{\texttt{ttest}}(g, D, \theta, \delta) - \tau \leq 0$, where each $\hat{H}_i$ implicitly depends on $\theta$ by the definition, $\hat{H}_i := \phi(T_i)h(X_i, Y_i, S_i, \theta)$. From (7), it is clear that if the pre-shift conditionals, $\Pr(\xi|T{=}t)$ and $\Pr(T{=}t|\xi)$, can be computed from the training data for all $t \in \mathcal{T}$, and if the post-shift demographic marginals, $\Pr(T'{=}t)$, are provided by the user during training, then a $(1{-}\delta)$-confidence upper bound $g'(\theta)$ can be computed even when data from the post-shift distribution is unavailable.

### 3.1.2 UNKNOWN DEMOGRAPHIC SHIFT

It is often unrealistic to assume that the post-shift marginal distribution is known exactly during training. To address this, we consider the setting in which the user provides a set, $\mathcal{Q} := \{(a_t, b_t)\}_{t \in \mathcal{T}}$, that contains non-empty intervals describing marginal distributions over $T'$ that might be encountered after deployment. Given $\mathcal{Q}$, we compute high-confidence upper bounds on $g'(\theta)$ by determining the largest value of the high-confidence upper bound attained for any $q \in \mathcal{Q}$.

First, we parameterize the high-confidence upper bound in (7) to explicitly depend on a particular choice of post-shift demographic distribution, $q$. Formally, we define $\hat{U}_{\texttt{ttest}}(g, D, \theta, \delta; q)$ by replacing all occurrences of $\phi(T_i)$ in (6) with $\phi(T_i; q)$, given by

$$\phi(t; q) := \frac{\Pr(\xi|T{=}t)q_t}{\Pr(T{=}t|\xi) \sum_{t' \in \mathcal{T}} \Pr(\xi|T{=}t')q_{t'}}, \tag{8}$$

where $q_t = \Pr(T'{=}t)$ for one possible demographic-shifted demographic distribution. While the true post-shift marginal distribution, $q^*$, is assumed to be unknown, it is clear that if $q^* \in \mathcal{Q}$, then $\hat{U}_{\texttt{ttest}}(g, D, \theta, \delta; q^*) \leq \sup_{q \in \mathcal{Q}} \hat{U}_{\texttt{ttest}}(g, D, \theta, \delta; q)$. It follows that,

$$\Pr\left(\mathbf{E}[H'|\xi'] \leq \sup_{q \in \mathcal{Q}} \hat{U}_{\texttt{ttest}}(g, D, \theta, \delta; q)\right) \geq \Pr\left(\mathbf{E}[H'|\xi'] \leq \hat{U}_{\texttt{ttest}}(g, D, \theta, \delta; q^*)\right) \geq 1{-}\delta.$$

Consequently, if $g'(\theta) := \mathbf{E}[H'|\xi'] - \tau$, then an algorithm, $a$, can be designed to satisfy Property B by following Algorithm 1 and defining the fairness test to only return models when $\sup_{q \in \mathcal{Q}} \hat{U}_{\texttt{ttest}}(g, D, \theta, \delta; q) - \tau \leq 0$. We propose to use a numerical optimizer to approximate the supremum of $\hat{U}_{\texttt{ttest}}$ over $q \in \mathcal{Q}$. In our implementations, we use *simplicial homology optimization* (Endres et al., 2018), which converges to the global optima of non-smooth functions subject to equality and inequality constraints such as those defined by the condition $q \in \mathcal{Q}$.

### 3.2 CANDIDATE SELECTION

Because it is ultimately the fairness test that causes `Shifty` to satisfy Property B, candidate selection can be implemented using any procedure for training a classifier without impacting the theoretical

provided by `Shifty`. However, it is advantageous to select a candidate model that appears to behave fairly. If model accuracy is correlated with unfair behavior, a candidate selection procedure that solely optimizes accuracy will tend to select models that will fail the fairness test. To mitigate this, we perform candidate selection by minimizing a loss consisting of two terms: one that estimates the worst-case classification error on the deployment distribution, and another that penalizes models that appear to be unfair. Specifically, if $\mathbb{I}[\cdot]$ denotes the indicator function that returns 1 if its argument is `true` and 0 otherwise, then the candidate model is found by minimizing,

$$\ell_{\texttt{Shifty-ttest}}(g, D_c, \theta, \delta; \mathcal{Q}) \coloneqq \texttt{Error}(D_c, \theta; \mathcal{Q}) + \texttt{Penalty}(g, D_c, \theta, \delta; \mathcal{Q}), \qquad (9)$$

where $\texttt{Error}(D_c, \theta; \mathcal{Q})$ estimates the wort case classification error using Theorem 1,

$$\texttt{Error}(D_c, \theta; \mathcal{Q}) \coloneqq \inf_{q \in \mathcal{Q}} \frac{1}{|D_c|} \sum_{(x,y,t) \in D_c} \mathbb{I}[\theta(x) \neq y] \, \phi(t),$$

and $\texttt{Penalty}(g, D_c, \theta, \delta; \mathcal{Q})$ penalizes models that are likely to be unfair after deployment,

$$\texttt{Penalty}(g, D_c, \theta, \delta; \mathcal{Q}) \coloneqq \max \left( 0, \sup_{q \in \mathcal{Q}} \hat{U}_{\texttt{ttest}}(g, D_c, \theta, \delta; q) \right).$$

## 4 EVALUATION

We evaluate the fairness and performance of `Shifty-ttest`, a `Shifty` algorithm based on the results derived in Section 3. Specifically, `Shifty-ttest` which is designed according to Algorithm 1, where the subroutine `TrainCandidate` minimizes (9), and `HighConfUB` computes $\sup_{q \in \mathcal{Q}} \hat{U}_{\texttt{ttest}}(g, D, \theta, \delta; q) - \tau$. Additional implementation details are provided in Appendix C.

Our evaluation answers three central research questions (RQ). **[RQ1]** In practice, do the models trained using `Shifty-ttest` or prior approaches adhere to high-probability fairness guarantees under demographic shift? **[RQ2]** Is `Shifty-ttest` able to train models whose accuracy is comparable to those produced by prior approaches that do not account for demographic shift? **[RQ3]** Does `Shifty-ttest` avoid returning NSF when reasonably sized training datasets are available?

Answering these questions requires *multiple* pairs of datasets sampled from the same distribution and exhibiting a consistent demographic shift. We generate multiple training and deployment datasets by resampling from a fixed population using known distributions. This ensures that failures—instances in which an algorithm returns unfair models with a larger frequency than guaranteed—are properly attributed to a failure of the algorithm instead of a violation of the user's assumptions on the demographic shift. In addition, knowledge of the training and deployment distributions can be used to compute exact values for accuracy, $g(\theta)$, and $g'(\theta)$ during evaluation. We uniformly sample training datasets from the population, train models using each algorithm, and evaluate fairness after deployment by using a new distribution over the population that satisfies the user's assumptions about the demographic shift. We conducted experiments using using two datasets, described below.

**UCI Adult Census Dataset:** This dataset includes various features, including race and sex, describing $48{,}842$ individuals taken during the 1994 US census (Kohavi and Becker, 1996). Specifically, we consider the subset of the dataset corresponding to black or white individuals. Using this dataset, we train classifiers to predict whether or not an individual earns above \$50,000 each year. To assess fairness under demographic shift, we define the fairness attribute, $S$, to be the race of each individual, and define the demographic shift to be over $T$, the sex of each individual.

**UFRGS Entrance Exam and GPA Dataset:** This dataset describes academic records for $43{,}303$ students from a university in Brazil (da Silva, 2019). For each student, the dataset includes a vector of entrance exam scores, a binary label representing if the student's GPA, between $0$ and $4.0$, was above $3.0$, and labels for the student's race and sex, which we treat as the demographic attribute and the fairness attribute, respectively. We discuss the results of these experiments in Appendix E.

We use `Shifty-ttest`, Seldonian algorithms, Fairlearn, Fairness Constraints, and RFLearn to enforce disparate impact and demographic parity constraints (see Appendix D.2). Importantly, while Fairlearn was not designed to avoid disparate impact, and Fairness Constraints was not designed to enforce disparate impact or demographic parity, we include them for illustrative purposes.

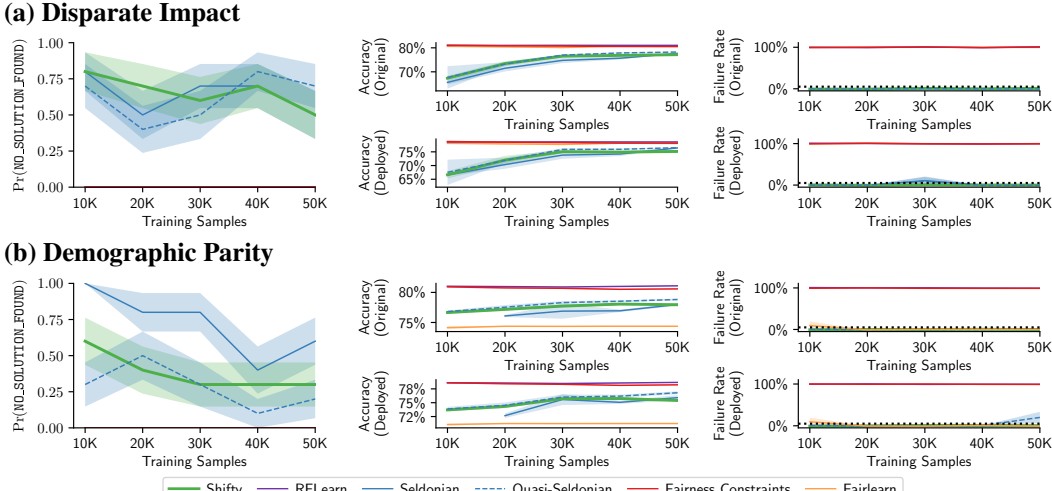

Figure 2: Results using the UCI Adult Census dataset under known demographic shift. For the experiments using disparate impact, RFLearn, Fairness Constraints, and Fairlearn all produce failure rates of 100% before and after deployment. In the experiments using demographic parity impact, RFLearn and Fairness Constraints both produce failure rates of 100% before and after deployment.

## 4.1 Experiments: Known Demographic Shift

To assess the impact of demographic shift on the fairness, we simulated a single illustrative example demographic shift shown in Figure 4a of Appendix D.3. For each experiment, we conducted 25 trials while varying the amount of training data in order to identify dependencies on the training dataset's size. For each trial, we uniformly sampled a training dataset and trained models using each algorithm. After training, we recorded whether each algorithm produced a model or NSF, as well as the average accuracy and prevalence of unfair behavior for each trained model both before and after deployment.

Figure 2 shows the results of these experiments using the UCI Adult Census dataset. For each definition of fairness, the leftmost plot shows the probability with which each algorithm returns NSF. To the right are two rows of plots showing evaluations of each algorithm on the training and deployment distributions, respectively. Within each row, the left plot displays the accuracy of models returned by each algorithm when trained using various amounts of data. The right plot in each row displays the frequency with which each algorithm returns an unfair model, which we call the failure rate. The horizontal dashed line shows the tolerance for unfair outcomes, $\delta=0.05$, set when training Shifty-ttest and Seldonian algorithms. Finally, shaded regions denote standard error bars.

Our experiments confirm that Shifty algorithms effectively avoid unfair behavior after demographic shift while prior algorithms do not. The failure rates of Shifty-ttest after demographic shift occurs (bottom right plots in Figure 2) are **always** below the tolerance set during training. However, while Seldonian algorithms were fair during training, they, along with the other baselines, frequently violate that fairness constraint after deployment. Interestingly, despite being designed to promote fairness with respect to demographic parity under covariate shift, RFLearn consistently violated the demographic parity constraints both before and after deployment. We found that the average value of $g'(\theta)$ for RFLearn models was 0.0175, which highlights the observation that many fair classification algorithms find models that are reasonably fair, but do not provide high-confidence guarantees.

Next, we found that Shifty-ttest provided guarantees of fairness with only a minor loss in accuracy compared to the baselines that do not provide any fairness guarantees. The plots in the middle column of Figure 2 show that Shifty-ttest produced models that achieved the same accuracy as those trained using standard Seldonian algorithms, with performance approaching that of the most accurate baselines for sufficiently large training datasets. This pattern is consistent with the results of our additional experiments in Appendix E.1, which show that for fairness constraints that are not excessively strict, Shifty-ttest returns models with competitive accuracy. Finally, we found that Shifty-ttest required slightly more training data than standard Seldonian algorithms to consistently avoid returning NSF (left plots in Figure 2).

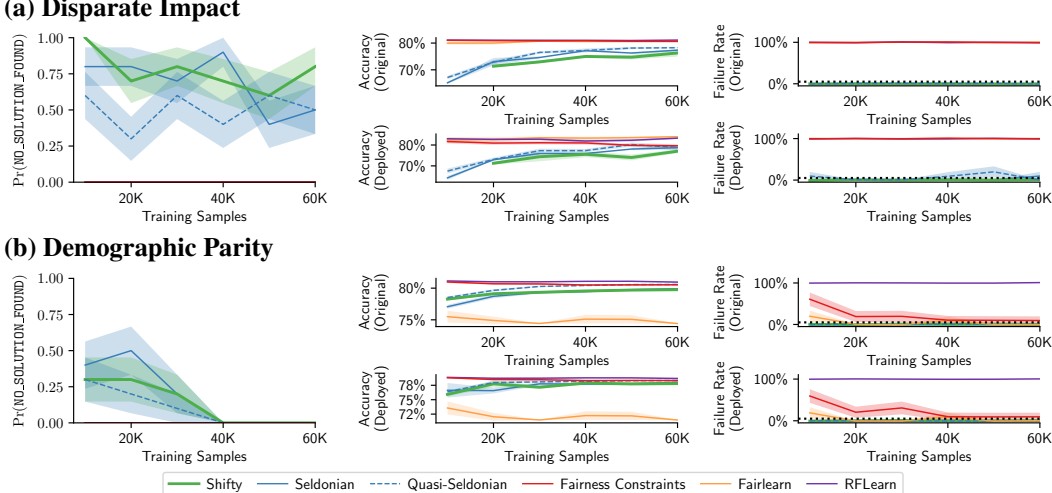

Figure 3: Results when enforcing fairness constraints using the UCI Adult Census dataset under unknown demographic shift. Note that for the experiments using disparate impact, RFLearn, Fairness Constraints, and Fairlearn all produce failure rates of 100% both before and after deployment.

## 4.2 EXPERIMENTS: UNKNOWN DEMOGRAPHIC SHIFT

Next, we conduct experiments assuming that the user has provided upper and lower bounds on the marginal probability of encountering individuals of each race after deployment instead of specifying the shift exactly. The example bounds used in our experiments are discussed in Appendix D.3. Using these bounds as the specification for demographic shift, we trained models using the same procedure as described in Section 4.1. To evaluate each model's performance after demographic shift, we performed a worst-case analysis and selected the deployment distribution over the population to satisfy the user's assumptions, but otherwise *maximize* the prevalence of unfair behavior when using that model to generate predictions (See Appendix D.1). As in Section 4.1, we report results for each experiment using five plots that show the probability of each algorithm returning `NSF`, as well as average accuracy and the failure rate of each algorithm before and after deployment.

Figure 3 shows results for disparate impact and demographic parity (additional results are provided in Appendix E), which confirm that `Shifty-ttest` provides guarantees of fair behavior that hold when the post-shift demographic distribution is not exactly known. While Seldonian algorithms are fair during training (Figure 3, top right), they are often unfair after demographic shift. Furthermore, `Shifty-ttest` generally produced models that achieved similar accuracy as those trained existing algorithms that do not satisfy Property B, although some fairness constraints were as harder to satisfy and caused `Shifty-ttest` to frequently return `NSF` (see Appendix E). These results indicate that `Shifty-ttest` is effective when fairness guarantees are critical, and is capable of training models that obtain comparable accuracy to those trained using unfair baselines.

## 5 CONCLUSION

We proposed `Shifty`, a strategy for designing classification algorithms that provide high-confidence fairness guarantees that hold when the distribution of demographics changes between training and deployment. This setting poses significant challenges for existing fair ML algorithms, as the fairness guarantees they provide generally assume a constant data distribution. In contrast, `Shifty` algorithms allow the proportions of demographics to change after training, provided the user has information describing this change. `Shifty` algorithms can be used when the new demographic proportions are known, or when these proportions are bounded in known intervals. Finally, we evaluated `Shifty-ttest`, an implementation of `Shifty` based on the Student's $t$-test, and found that the fairness guarantees it provides are empirically valid, wherein models trained using existing fair algorithms consistently produced unfair outcomes.

## ACKNOWLEDGMENTS

This work is supported by the National Science Foundation under grants no. CCF-1763423 and CCF-2018372, by the DEVCOM Army Research Laboratory under Cooperative Agreement W911NF-17-2-0196 (ARL IoBT CRA), and by Adobe, Google Research, Kosa.ai, and Meta Research.

## ETHICS STATEMENT

The primary goal of this research is to identify and overcome practical challenges, namely demographic shift, that might cause current algorithms to produce unfair outcomes. Our contributions provide tools needed for data scientists and ML practitioners to use ML in conscientious, ethical ways. However, we note that, when applying algorithms such as Shifty, it is important to carefully select the definition of unfair behavior to be appropriate for the problem at hand. While we evaluate Shifty using five standard definitions of unfair behavior for illustration, many definitions have been proposed and studied (Verma and Rubin, 2018), some of which cannot be simultaneously enforced (Chouldechova, 2017; Corbett-Davies et al., 2017; Kleinberg et al., 2017). Consequently, while Shifty offers a valuable tool for enforcing fairness constraints, ML practitioners should carefully study their target application, ideally working with domain experts and stakeholders, to ensure that the definitions they select meaningfully capture the unfair behaviors they wish to avoid.

Next, we note that the fairness guarantees provided by Shifty may fail to hold if one or more assumptions made by the algorithm do not hold. Most notably, if the demographic shift assumptions, (2) and (3), do not hold for the user's choice of demographic attribute $T$, then the guarantees provided by Shifty may not hold in practice. Similarly, if the user's specification of possible demographic shift, defined by the input $\mathcal{Q}$, does not accurately represent the demographic shift that occurs, then the guarantees provided by our algorithms may be invalidated. Thus, it is important that ML practitioners study the application domain in order to ensure that the inputs they supply to Shifty are appropriate.

Finally, Shifty-ttest is based on inversion of the Student's $t$-test, which only holds exactly if the observations, $H_i$ are not normal. While this approximation error becomes smaller as more samples are used for training, the high-confidence guarantees provided by Shifty may not hold with $1-\delta$ probability when trained on very small datasets. Therefore, in applications for which very few training observations are available, ML practitioners should employ Shifty algorithms based on other, non-approximate confidence intervals.

## REPRODUCIBILITY STATEMENT

To support efforts to reproduce our results, all code and data used in this paper will be made publicly available upon publication. Proofs of our theoretical results can be found in Appendix B and implementation details for our algorithms can be found in Section 3 and Appendix C. In addition, experimental details can be found in Section 4, full descriptions of the fairness definitions we tested are shown in Appendix D.2, and additional experimental details and results are included in Appendix D and Appendix E, respectively. The UCI Adult Census dataset is available for download from the UCI Machine Learning Repository (Kohavi and Becker, 1996), and the UFRGS Entrance Exam and GPA dataset is available at `https://doi.org/10.7910/DVN/O35FW8` (da Silva, 2019). Code for reproducing our experiments can be found at `https://github.com/sgiguere/Fairness-Guarantees-under-Demographic-Shift`, which includes instructions for adding the UCI Adult Census and UFRGS GPA datasets to the codebase.

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

## A  ALTERNATIVE APPROACHES FOR ACHIEVING FAIR OUTCOMES

Designing algorithms that meet fairness requirements with high-confidence can be seen as a type of constrained optimization problem. Algorithms that enforce *hard constraints* search for a predictive model within a feasible set of models satisfying the user's fairness constraints (Irani, 2015). Soft constraints on the objective function used to guide the search for a model can achieve fairness empirically (Zafar et al., 2017), and multi-objective methods can be used to satisfy multiple, potentially conflicting fairness objectives (Smits and Kotanchek, 2005). Unfortunately, algorithms based on hard or soft constraints alone do not provide high-confidence fairness guarantees. Fair algorithms can be designed based on *chance-constrained programming* (Charnes and Cooper, 1959; Miller and Wagner, 1965; Prékopa, 1970), in which an objective function is optimized subject to a set of constraints. This formulation can provide high-confidence fairness guarantees but typically requires knowledge of the distribution of each variable used to quantify fairness, which is often impractical. In contrast, our methods provide assurances of fair behavior without these limiting assumptions.

In our work, we consider the problem of establishing high-confidence fairness guarantees that hold under demographic shift, specifically when no samples are available from the deployment distribution. While `Shifty` is, to our knowledge, the first to provide high-confidence guarantees of fairness in this setting, there are several existing strategies for promoting fair outcomes under various forms of distribution shift, which leverage a variety of assumptions. For example, approaches have been proposed to promote fair outcomes under *concept shift*, which occurs when the distribution over safety attributes and feature vectors changes between training and deployment. Schumann et al. (2019) derive bounds describing how well fairness properties can be transferred from a source to a target domain, and use them to propose training procedures that can improve the transferability of fairness properties. While this approach could be applied to overcome demographic shift, it does not provide guarantees that the resulting models will meet specific fairness tolerances, and it assumes that data is available from the deployment distribution and therefore cannot be applied in the setting we consider. In contrast to methods for overcoming concept shift, many works on promoting fairness under distribution shift assume that the shift has a particular structure. For example, Dai (2020) propose a fairness-aware model for *label shift* and *label bias*, which occur when the distribution over the fairness attribute and feature vector is the same during training and after deployment, but the conditional distribution over true labels differs. Similarly, Biswas and Mukherjee (2021) propose an algorithm for promoting fair outcomes under *prior probability shift*, which occurs when the marginal distribution of true labels changes between training and deployment, using *proportional equality* to measure fairness. Unfortunately, the assumptions of label shift, label bias, and proportional equality are contrary to the assumptions of demographic shift, so that these approaches cannot be applied to the problem setting we consider.

Among the problem settings considered in prior work, the most similar setting to demographic shift is *covariate shift*, which occurs when the distribution of fairness attributes and feature vectors changes between training and deployment, but the conditional distribution over labels is unchanged. Indeed, if, in addition to the demographic shift assumptions, the user assumes that the distribution of true labels is conditionally independent of the demographic attribute given the features and safety attributes,

then demographic shift can be viewed as a form of covariate shift. Several approaches have been proposed for promoting fair outcomes under covariate shift and various definitions of fairness, but these approaches cannot be applied to our problem setting because they make the assumption that samples are available from the deployment distribution. Furthermore, these approaches are not proven to satisfy Property B (Rezaei et al., 2021; Coston et al., 2019). Singh et al. (2021) propose a training algorithm that ensures that fairness properties of the resulting model are invariant to covariate shift by exploiting a causal graph of the problem that is provided by the user. While this approach can be applied without samples from the deployment distribution, the causal graph required by the algorithm is often unknown, and the algorithm does not provide fairness guarantees. Finally, Du and Wu (2021) propose an algorithm, RFLearn, that promotes fair outcomes under covariate shift without access to data from the deployment environment. While this approach does not provide fairness guarantees, and despite the differences between demographic shift and covariate shift, we include RFLearn in our experimental designs for illustration.

We propose algorithms that produce fair models following the design principles outlined by Seldonian ML (Thomas et al., 2019) while accounting for demographic shift. Other ways to account for demographic shift include augmenting training data with synthetically generated variations or antagonistic examples, or explicitly regularizing the training objective using unlabeled data from the deployment distribution, which improves the models' generalization (Goodfellow et al., 2016; Zhang and LeCun, 2017). However, these approaches do not tackle fairness and cannot be directly applied to provide fairness guarantees under demographic shift. If the change in distribution can be addressed by a transformation of either the features or response variables, methods can learn this transformation using access to data from the deployment distribution combined with assumptions, such as that the transformation is linear (Fernando et al., 2014; Gong et al., 2012; Gopalan et al., 2011), or by iteratively assigning predicted labels to unlabeled data from the deployment environment (Bruzzone and Marconcini, 2009). However, such approaches are ill-suited for establishing high-confidence fairness guarantees under demographic shift.

Approaches can account for the differences between the training and deployment environments by reweighting the contribution of each training observation according to the relative probability of encountering that observation. Such methods are effective at improving performance in classification (Zadrozny, 2004; Huang et al., 2006), density estimation (Dudík et al., 2006), and regression (Huang et al., 2006). While most of these approaches focus on accuracy, some have focused on safety constraints, such as fairness. Lipton et al. (2018) proposed correcting for covariate shift by using a reweighting scheme to compute intervals on a classifier's confusion rates and applying a correction step based on these intervals. This approach produces empirically fair models but does not provide high-confidence guarantees. An algorithm's fairness can be improved by manipulating the underlying data, e.g., by removing data that violates fairness properties (Verma et al., 2021) or inserting data inferred using fairness properties (Salimi et al., 2019). These methods, however, do not provide guarantees. Fairness can also be enforced in post-processing (Awasthi et al., 2020) but, again, without guarantees.

## B  PROOF OF THEOREM 1

**Theorem 1.** *Assume that* $\Pr(T{=}t) \geq 0$ *for all* $t \in \mathcal{T}$. *If the demographic shift properties hold, then the random variable* $\hat{H} := \phi(T)H$ *satisfies* $\mathbf{E}[H'|\xi'] = \mathbf{E}[\hat{H}|\xi]$, *where* $\phi$ *is defined by (5).*

*Proof.* First, we write $\mathbf{E}[H'|\xi']$ as a sum over expected values conditioned on the value of the demographic attribute by applying the law of total probability (Zwillinger and Kokoska, 1999):

$$\mathbf{E}[H'|\xi'] = \sum_{t \in \mathcal{T}} \mathbf{E}[H'|\xi', T'{=}t] \Pr(T'{=}t|\xi')$$

$$= \sum_{t \in \mathcal{T}} \mathbf{E}[H|\xi, T{=}t] \Pr(T'{=}t|\xi'). \qquad \text{(Using (3))}$$

Here, the second line follows from the second demographic shift assumption, which states that for all $t \in \mathcal{T}$ and $x, y, s \in \mathcal{X} \times \mathcal{Y} \times \mathcal{S}$, $\Pr(X'{=}x, Y'{=}y, S'{=}s|T'{=}t) = \Pr(X{=}x, Y{=}y, S{=}s|T{=}t)$. Next, we multiply each term by $\Pr(T{=}t|\xi)/\Pr(T{=}t|\xi) = 1$, reorganize terms, and write the sum

over $t \in \mathcal{T}$ as a single expected value:

$$
\begin{aligned}
\mathbf{E}[H'|\xi] &= \sum_{t \in \mathcal{T}} \mathbf{E}[H|\xi, T{=}t] \left( \frac{\Pr(T{=}t|\xi)}{\Pr(T{=}t|\xi)} \right) \Pr(T'{=}t|\xi') \\
&= \sum_{t \in \mathcal{T}} \mathbf{E}[H|\xi, T{=}t] \left( \frac{\Pr(T'{=}t|\xi')}{\Pr(T{=}t|\xi)} \right) \Pr(T{=}t|\xi) \\
&= \mathbf{E}\left[ \phi(T)H|\xi \right],
\end{aligned}
$$

where

$$
\phi(t) = \frac{\Pr(T'{=}t|\xi')}{\Pr(T{=}t|\xi)}.
$$

Finally, we rewrite $\phi(t)$ to depend on the post-shift marginal distribution, $\Pr(T'{=}t)$, and the pre-shift conditional distributions, $\Pr(T{=}t|\xi)$ and $\Pr(C|T{=}t)$, for each $t \in \mathcal{T}$:

$$
\begin{aligned}
\phi(t) &= \frac{\Pr(T'{=}t|\xi')}{\Pr(T{=}t|\xi)} \\
&= \frac{\Pr(\xi'|T'{=}t)\Pr(T'{=}t)}{\Pr(T{=}t|\xi)\Pr(\xi')} && \textit{(Using Bayes' Theorem)} \\
&= \frac{\Pr(\xi|T{=}t)\Pr(T'{=}t)}{\Pr(T{=}t|\xi)\Pr(\xi')} && \textit{(Using (3))} \\
&= \frac{\Pr(\xi|T{=}t)\Pr(T'{=}t)}{\Pr(T{=}t|\xi)\sum_{t' \in \mathcal{T}}\Pr(\xi'|T'{=}t')\Pr(T'{=}t')} \\
&= \frac{\Pr(\xi|T{=}t)\Pr(T'{=}t)}{\Pr(T{=}t|\xi)\sum_{t' \in \mathcal{T}}\Pr(\xi|T{=}t')\Pr(T'{=}t')}. && \textit{(Using (3))}
\end{aligned}
$$

$\square$

# C   ADDITIONAL IMPLEMENTATION DETAILS FOR SHIFTY

## C.1   DATA PARTITIONING

The data partitioning step ensures that the outcome of the candidate selection step is independent of the outcome of the fairness test, which is necessary to guarantee that the overall algorithm satisfies (1) under demographic shift.

To illustrate the requirement for this step, consider an algorithm that accepts a set of training observations, $D$, uses $D$ to select a candidate model, $\theta_c$, and finally uses $D$ again to evaluate the high-confidence upper bound on $g'(\theta_c)$ to perform a fairness test. While this algorithm might appear to be fair, it is not guaranteed to satisfy Property B because the output of candidate selection, $\theta_c$, is correlated with the outcome of the fairness test because the same input data, $D$, is used to perform both steps. Consequently, it is possible that candidate selection can consistently select models that cause the high-confidence upper bound used by the fairness test to under-estimate the value of $g'(\theta_c)$. By ensuring that candidate selection and the fairness test use independent sets of observations, this correlation is eliminated, causing the probability of the high-confidence upper bound failing to be no more than $\delta$ as required by Property B.

Finally, we note that one area of future research might consider the optimal way to split an input dataset into parts used for candidate selection, $D_c$, and for evaluating the fairness test, $D_f$. Specifically, increasing the size of $D_c$ improves the ability of the candidate selection step to identify models that are accurate and generalize well, but reduces the size of $D_f$ and makes the fairness test more difficult to pass. In our experiments, we split the input data evenly between $D_c$ and $D_f$, but we hypothesize that there may be more effective techniques for determining the optimal splitting proportion.

## C.2 ENFORCING COMPLEX FAIRNESS DEFINITIONS

In Section 3, we assume that the user-specified definition of unfair behavior, $g$, is defined by,

$$g(\theta) \coloneqq \mathbf{E}\big[\, H \,\big|\, \xi \,\big] - \tau. \tag{10}$$

However common fairness definitions such as the $80\%$ rule (Griggs v. Duke Power Co., 1971) and equalized odds (Hardt et al., 2016) do not have the same form as (10). Nonetheless, these definitions can often be described by known *expressions* of multiple terms, where each term is a conditional or marginal expected value. Consider, for example, the definition of bias codified by *disparate impact* (Griggs v. Duke Power Co., 1971; Chouldechova, 2017; Zafar et al., 2017):

$$g_{\mathrm{DI}}(\theta) \coloneqq 0.8 - \min\left\{ \frac{\mathbf{E}[\theta(X)|S{=}\texttt{female}]}{\mathbf{E}[\theta(X)|S{=}\texttt{male}]}, \frac{\mathbf{E}[\theta(X)|S{=}\texttt{male}]}{\mathbf{E}[\theta(X)|S{=}\texttt{female}]} \right\}.$$

This definition is clearly not of the form described by (10); furthermore, it is challenging to estimate because i.i.d. samples of $\mathbf{E}[\theta(X)|S{=}\texttt{male}]$ and $\mathbf{E}[\theta(X)|S{=}\texttt{female}]$ cannot be simultaneously computed given a single observation. Regardless, it is possible to compute valid high-confidence upper bounds on $g_{\mathrm{DI}}(\theta)$ by leveraging the recursive bound-propagation described by Metevier et al. (2019). Importantly, we evaluate our proposed algorithms using complex but realistic fairness definitions by leveraging this strategy, and as a consequence, our results are influenced by this decision.

While we refer the reader to Metevier et al. (2019) for a more complete discussion of this strategy, we present a brief intuition here. First, we assume that the user's definition of $g$ can be written as,

$$g(\theta) = f\big(\phi_1(\theta), ..., \phi_k(\theta)\big),$$

where each $\phi_i$ for $i \in \{1, ..., k\}$ denotes a *parameter* of the joint distribution of $(X, Y, S)$, and $f$ is some function of $k$ arguments specified by the user. In our work, we assume that these parameters are each expressed as a conditional expected value analogous to (10) and that the expression defining $f$ is provided by the user as text.

Following (Metevier et al., 2019), the expression for $f$ is parsed into a tree structure representing the recursive application of various predefined operations. To construct a high-confidence upper bound on $g(\theta)$—or in this work, $g'(\theta)$—we first construct a set of confidence intervals on each parameter using the methods described in Section 3. Importantly, we apply the union bound to ensure that the set of confidence intervals on the parameters hold jointly with probability $1-\delta$. Next, these intervals are recursively propagated through the expression for $f$, where at each node of the computation tree, the interval for that node is computed by applying *interval arithmetic*, which describes the image of certain mathematical operations given intervals as their arguments. Since the root of the computation tree denotes the quantity $g'(\theta)$, the result of this recursive system is an interval containing the true value of $g'(\theta)$ with at least probability $1-\delta$.

A drawback of this approach is that it assumes that the intervals describing each parameter are independent, which is often false when considering demographic shift. However, it is important to note that violation of this assumption does not impact the validity of the resulting confidence interval on $g'(\theta)$. Instead, when the input parameters are not dependent, the resulting confidence interval on $g'(\theta)$ is larger than it would be if the dependence between each parameter were known and accounted for. For example, suppose $g$ is defined to measure bias based on sex, and consider two races. If, for individuals of one race, a certain sex is encountered much more often than other sexes, while for individuals of the second race, all sexes are encountered equally often, then a demographic shift that makes the first race more likely may cause certain parameters defining fairness to be highly correlated with others. By ignoring these dependencies, the approach presented in (Metevier et al., 2019) may produce significantly larger confidence intervals for $g'(\theta)$ compared to alternative approaches that leverage this dependency. For this reason, we consider this problem to be a strong candidate for future work, as it has the potential to improve the data efficiency of our methods as well as those proposed in existing work (Metevier et al., 2019; Thomas et al., 2019).

# D    ADDITIONAL EXPERIMENTAL DESIGN DETAILS

## D.1    A STRATEGY FOR SIMULATING AND EVALUATING DEMOGRAPHIC SHIFT

Here, we describe our procedure for simulating the impact of demographic shift given a fixed population dataset when the exact deployment distribution is unknown. Intuitively, after generating a training dataset, we antagonistically select a new, non-uniform distribution over the population that satisfies the user's demographic shift assumptions—that is, that the marginal distribution over demographics is contained in $\mathcal{Q}$—but otherwise maximizes the prevalence of unfair behavior. Since the population and sampling distributions are known during evaluation, this oracle knowledge can be used to compute exact values for various statistics, such as expected classification accuracy. To make this procedure formal, let the population dataset be denoted by $\mathcal{D}_{pop}$:

$$\mathcal{D}_{pop} := \{(x_i, y_i, s_i, t_i)\}_{i=1}^n.$$

Note that we do not refer to this set using the standard notation for random variables because in our experimental context the population is treated as a fixed, non-random population. To generate a random training dataset, $D$, we sample observations uniformly from $\mathcal{D}_{pop}$ with replacement. Specifically, if $P$ denotes the uniform distribution over the observations in $\mathcal{D}_{pop}$, then training datasets are defined by $D := \{(X_j, Y_j, S_j, T_j)\}_{j=1}^{n_0}$, where each $(X_j, Y_j, S_j, T_j) \sim P$.

Next, we generate a new distribution over the population that satisfies the user's assumptions but otherwise maximizes the prevalence of unfair behavior for a given model, which we denote by $Q$. However, to comply with the user's assumptions about the demographic shift, $Q$ must be selected carefully. The following theorem provides the conditions that $Q$ must satisfy to achieve this.

**Theorem 2.** *Let $P$ denote a uniform distribution over $\mathcal{D}_{pop} := \{(x_i, y_i, s_i, t_i)\}_{i=1}^n$. Assume that the demographic attribute takes values in some set $\mathcal{T}$ and that each demographic $t \in \mathcal{T}$ occurs at least once in the population. Next, let each $q \in \mathcal{Q}$ denote a marginal distribution over $\mathcal{T}$, where $q_t$ denotes the probability of encountering demographic $t$. Finally, let $\mathbb{N}_{\mathcal{D}_{pop}}[x, y, s, t]$ denote the number of observations in $\mathcal{D}_{pop}$ that are equal to $(x, y, s, t)$ and let $\mathbb{N}_{\mathcal{D}_{pop}}[t]$ denote the number of observations that have demographic attribute equal to $t$. It follows that $Q$, defined below, is a distribution over $\mathcal{D}_{pop}$ that satisfies both of the demographic shift assumptions, and has a marginal distribution over demographics that is contained in $\mathcal{Q}$:*

$$Q(X{=}x, Y{=}y, S{=}s, T{=}t) = \frac{\mathbb{N}_{\mathcal{D}_{pop}}[x, y, s, t]}{\mathbb{N}_{\mathcal{D}_{pop}}[t]} q_t.$$

*Proof.* To show this result, we derive an expression for $Q$ that has these properties by construction.

First, we expand the post-shift joint distribution using the laws of conditional probability:

$$Q(X{=}x, Y{=}y, S{=}s, T{=}t) = Q(X{=}x, Y{=}y, S{=}s|T{=}t)Q(T{=}t).$$

Next, we apply the second demographic shift assumption:

$$Q(X{=}x, Y{=}y, S{=}s, T{=}t) = P(X{=}x, Y{=}y, S{=}s|T{=}t)Q(T{=}t).$$

Then, we represent the conditional $P(X, Y, S|T{=}t)$ as a ratio using laws of conditional probability:

$$Q(X{=}x, Y{=}y, S{=}s, T{=}t) = \frac{P(X{=}x, Y{=}y, S{=}s, T{=}t)}{P(T{=}t)} Q(T{=}t).$$

Because $P$ is a uniform distribution over $\mathcal{D}_{pop}$, it follows that the value of $P(X{=}x, Y{=}y, S{=}s, T{=}t)$ is simply the number of occurrences of $(x, y, s, t)$ in $\mathcal{P}$ divided by the total number of samples in the population, $n$. Similarly, $P(T{=}t)$ is equal to the number of observations that have demographic attribute equal to $t$, divided by $n$. Since we assume that each demographic is observed in the population, it follows that $P(T{=}t) > 0$ for all $t \in \mathcal{T}$. Let $\mathbb{N}_{\mathcal{D}_{pop}}[x, y, s, t]$ denote the number of observations in $\mathcal{D}_{pop}$ equal to $(x, y, s, t)$ and let $\mathbb{N}_{\mathcal{D}_{pop}}[t]$ denote the number of observations that have demographic attribute equal to $t$. It follows that for all observations, $(x, y, s, t) \in \mathcal{D}_{pop}$, we have

$$Q(X{=}x, Y{=}y, S{=}s, T{=}t) = \frac{\mathbb{N}_{\mathcal{D}_{pop}}[x, y, s, t]}{\mathbb{N}_{\mathcal{D}_{pop}}[t]} Q(T{=}t).$$

Finally, we define the marginal distribution of $Q$ over demographics to be given by $q$:

$$Q(X{=}x, Y{=}y, S{=}s, T{=}t) = \frac{\mathbb{N}_{\mathcal{D}_{pop}}[x, y, s, t]}{\mathbb{N}_{\mathcal{D}_{pop}}[t]} q_t.$$

Since $Q$ has the same conditional distribution given the demographic as $P$ by construction, it satisfies the demographic shift assumptions. Furthermore, since the marginal distribution of $Q$ over demographics is defined to be given by $q$, it also satisfies the user's assumptions. $\qquad\square$

Theorem 2 shows that, given a $q \in \mathcal{Q}$, it is straightforward to construct a distribution over the population which satisfies the demographic shift assumptions. Therefore, to select a distribution that maximizes the prevalence of unfair behavior for a given model, $\theta$, we numerically optimize $g'(\theta)$ over $q \in \mathcal{Q}$ using simplical homology optimization (Endres et al., 2018) to determine the maximizing marginal distribution, $q^*$, and then define the final distribution over $\mathcal{D}_{\text{pop}}$ using Theorem 2.

Theorem 2 can also be used to compute exact values for various statistics of interest during evaluation, such as expected classification accuracy or the value of $g'(\theta)$. Consider estimating the post-deployment classification accuracy of a model, $\theta$, given by $\mathbf{E}_Q[\mathbb{I}[\theta(X){=}Y]]$. If $\bar{\mathcal{D}}_{pop}$ denotes the set of unique observations in $\mathcal{D}_{pop}$, then we have

$$\begin{aligned}
&\mathbf{E}_Q\big[\,\mathbb{I}[\theta(X){=}Y]\,\big] \\
&= \sum_{(x,y,s,t) \in \bar{\mathcal{D}}_{pop}} \mathbb{I}[\theta(x){=}y]\, Q(X{=}x, Y{=}y, S{=}s, T{=}t) \\
&= \sum_{(x,y,s,t) \in \bar{\mathcal{D}}_{pop}} \mathbb{I}[\theta(x){=}y] \frac{\mathbb{N}_{\mathcal{D}_{pop}}[x,y,s,t]}{\mathbb{N}_{\mathcal{D}_{pop}}[t]}\, Q(T{=}t).
\end{aligned}$$

## D.2 FAIRNESS DEFINITIONS USED IN EXPERIMENTS

We evaluate the performance and fairness of various algorithms for several definitions of fairness, shown below. These definitions were specified as text input to `Shifty` and were bounded using a recursive technique used by prior Seldonian algorithms (Metevier et al., 2019) (See Appendix C.2).

**Demographic Parity (Dwork et al., 2012; Calders and Verwer, 2010):**

$$g_{\text{DP}}(\theta) := \Big| \mathbf{E}[\theta(X)|S{=}s_0] - \mathbf{E}[\theta(X)|S{=}s_1] \Big| - \epsilon_{\text{DP}}$$

**Disparate Impact (Griggs v. Duke Power Co., 1971; Chouldechova, 2017; Zafar et al., 2017):**

$$g_{\text{DI}}(\theta) := \epsilon_{\text{DI}} - \min\left\{ \frac{\mathbf{E}[\theta(X)|S{=}s_0]}{\mathbf{E}[\theta(X)|S{=}s_1]}, \frac{\mathbf{E}[\theta(X)|S{=}s_1]}{\mathbf{E}[\theta(X)|S{=}s_0]} \right\}$$

**Equal Opportunity (Hardt et al., 2016; Chouldechova, 2017):**

$$g_{\text{EOp}}(\theta) := \Big| \mathbf{E}[\theta(X)|Y{=}0, S{=}s_0] - \mathbf{E}[\theta(X)|Y{=}0, S{=}s_1] \Big| - \epsilon_{\text{EOp}}$$

**Equalized Odds (Hardt et al., 2016):**

$$\begin{aligned}
g_{\text{EOd}}(\theta) := \Big| &\mathbf{E}[\theta(X)|Y{=}0, S{=}s_0] - \mathbf{E}[\theta(X)|Y{=}0, S{=}s_1] \Big| + \\
&\mathbf{E}[1{-}\theta(X)|Y{=}1, S{=}s_0] - \mathbf{E}[1{-}\theta(X)|Y{=}1, S{=}s_1] \Big| - \epsilon_{\text{EOd}}
\end{aligned}$$

**Predictive Equality (Chouldechova, 2017; Corbett-Davies et al., 2017):**

$$g_{\text{PE}}(\theta) := \Big| \mathbf{E}[1{-}\theta(X)|Y{=}1, S{=}s_0] - \mathbf{E}[1{-}\theta(X)|Y{=}1, S{=}s_1] \Big| - \epsilon_{\text{PE}}$$

| Sex | $\Pr(T{=}t)$ | $\Pr(T'{=}t)$ | $\mathcal{Q}$ |
|---|---|---|---|
| $sex_1$ | 0.324 | 0.150 | (0.162, 0.661) |
| $sex_2$ | 0.676 | 0.850 | (0.338, 0.838) |

| Race | $\Pr(T{=}t)$ | $\Pr(T'{=}t)$ | $\mathcal{Q}$ |
|---|---|---|---|
| $race_1$ | 0.006 | 0.300 | (0.004, 0.304) |
| $race_2$ | 0.871 | 0.600 | (0.610, 0.909) |
| $race_3$ | 0.054 | 0.050 | (0.038, 0.338) |
| $race_4$ | 0.067 | 0.048 | (0.047, 0.347) |
| $race_5$ | 0.002 | 0.002 | (0.002, 0.301) |

(a) Marginal distributions of the demographic attribute for the UCI Adult Census experiments. The demographic attribute represents the anonymized sex.

(b) Marginal distributions of the demographic attribute for the UCI Adult Census experiments. The demographic attribute represents anonymized student race.

Figure 4: Marginal distributions over the demographic attribute, $T$, for the UCI Adult Census experiments (left) and the UFRGS GPA experiments (right). In each table, the left column shows the distribution of the demographic attribute during training, the middle column shows the deployed marginal distribution used in our experiments for known demographic shift (Section 4.1), and the right column shows the bounds on the marginal distribution used in our experiments for unknown demographic shift (Section 4.2).

For both the UCI Adult Census dataset and the UFRGS GPA dataset, we conducted separate experiments for each definition. For both sets of experiments, the fairness attribute, $S$, was allowed to take one of two possible values, $s_0$ or $s_1$. Specifically, in our experiments using the UCI Adult Census dataset, $S$ represents the race of each individual (i.e., $s_0 \coloneqq \texttt{black}$ and $s_1 \coloneqq \texttt{white}$) whereas in our experiments using the UFRGS GPA dataset (i.e., $s_0 \coloneqq \texttt{female}$ and $s_1 \coloneqq \texttt{male}$). Finally, the tolerances for each definition of fairness were set to $\epsilon_{\text{DP}} = 0.05$, $\epsilon_{\text{DI}} = 0.8$, $\epsilon_{\text{EOp}} = 0.15$, $\epsilon_{\text{EOd}} = 0.2$ and $\epsilon_{\text{PE}} = 0.025$ for our experiments using the UCI Adult Census dataset, and $\epsilon_{\text{DP}} = 0.1$, $\epsilon_{\text{DI}} = 0.8$, $\epsilon_{\text{EOp}} = 0.05$, $\epsilon_{\text{EOd}} = 0.1$ and $\epsilon_{\text{PE}} = 0.05$ for our experiments using the UFRGS GPA dataset. While the choice of tolerance for each definition impacts how strict the resulting fairness constraint it, we note that `Shifty` algorithms will satisfy their high-confidence fairness guarantees regardless of how these tolerances are chosen. The tolerances used in our experiments are for illustrative purposes, and in practice, will be set based on the specific requirements of the user's application.

### D.3 Simulated Demographic Shifts Used in Experiments

In our evaluations, we conduct separate experiments using the UCI Adult Census dataset and the UFRGS GPA dataset. In addition, for each dataset, we conduct experiments testing a single, known demographic shift, as well as an unknown demographic shift described by a set $\mathcal{Q}$ as described in Section 3.1.2. Here we discuss the specific simulated demographic shifts for each experiment.

Figure 4 describes the marginal distributions over the demographic attribute $T$ used for each experiment. Figure 4a provides the marginal distributions used in our experiments using the UCI Adult Census dataset, while Figure 4b provides the marginal distributions used in our experiments using the UFRGS GPA dataset. Within each table, the leftmost column shows the marginal probabilities of each value of $T$ during training. Next, the middle column of each table shows the distribution of the demographic attribute during deployment, which we simulate using the procedure described in Appendix D.1. Finally, the rightmost column of each table shows the bounds on the marginal distribution of the demographic attribute for each experiment that we use as a specification of $\mathcal{Q}$ in our experiments on overcoming unknown demographic shift. We generated these bounds by interpolating between the interval $(0, 1)$ and the singleton interval containing the marginal distribution over race under the training distribution, using an interpolation factor of $0.25$ for our experiments using the UCI Adult Census dataset, and $0.3$ for our experiments using the UFRGS GPA dataset. We note that this interpolation scheme and the choice of interpolation factors were chosen for illustrative purposes in our experiments, while in practice, $\mathcal{Q}$ will be specified by the user based on domain knowledge that is available during training.

### E  Additional Experimental Results

In this section, we present results for additional experiments conducted using the UCI Adult Census dataset (Kohavi and Becker, 1996) and the UFRGS GPA dataset (da Silva, 2019) to evaluate the

effectiveness of `Shifty-ttest` and various baseline algorithms at enforcing fairness constraints under demographic shift. While Section 4.1 provides results of our experiments using the UCI Adult Census dataset for enforcing disparate impact and demographic parity constraints, this section augments these results in two ways. First, for both known and unknown demographic shifts, we provide results of experiments using the UCI Adult Census dataset to enforce fairness constraints based on the concepts of equal opportunity, equalized odds, and predictive equality. Second, for both known and unknown demographic shifts, we provide results for experiments based on enforcing these same definitions using the UFRGS GPA dataset. Similarly to Section 4, we first present result for known demographic shift in Section E.1, and then provide results for our experiments using unknown demographic shift in Section E.2.

## E.1 KNOWN DEMOGRAPHIC SHIFT

Additional results for experiments using the UCI Adult Census dataset are shown in Figure 5, and full results for the experiments using the UFRGS GPA dataset are shown in Figure 6. First, we found that in all of our experiments, `Shifty-ttest` satisfied Property B, as evidenced by the fact that the green curves in the bottom rightmost plots of Figure 5 and Figure 6 are never above the dashed black line that denotes the $\delta = 0.05$ tolerance. On the other hand, Fairlearn, Fairness Constraints, and RFLearn all produced unfair models frequently, with some exceptions. For example, in our experiments using the UCI Adult Census dataset, Fairlearn was consistently fair when enforcing equal opportunity constraints (Figure 5, top right). In other experiments, such as our experiments using the UFRGS GPA dataset to enforce disparate impact, demographic parity, equalized odds, and predictive equality, we found that RFLearn and Fairlearn were often fair empirically, as evidenced by failure rates close to the $\delta = 0.05$ tolerance, but frequently exceeded the $0.05$ threshold after deployment. These findings show that, while existing fair algorithms can be effective at training models that are fair, they do not provide fairness guarantees that hold under demographic shift. Therefore a user cannot assume that the models produced by these methods will be fair in practice.

Next, our results using the UCI Adult Census dataset show that `Shifty-ttest` attained competitive accuracy compared to the best-performing baseline algorithms (middle column plots in Figure 5).

Consequently, `Shifty-ttest` suffered very little loss in accuracy in order to enforce fairness constraints that hold under demographic shift for this experiment. Results were comparable in our experiments using the UFRGS GPA dataset, with the exception that, when enforcing equal opportunity, equalized odds and predictive equality with the tolerances described in Appendix D.2, the resulting constraints were too strict for `Shifty-ttest` to reliably return models that are fair after demographic shift. Therefore, we conclude that, if suitably fair models exist, `Shifty-ttest` is effective at them when provided with sufficient training data, and that in many cases the models found by `Shifty` have comparable accuracy to those found by algorithms that do not provide fairness guarantees.

Finally, we note that `Shifty-ttest` was typically able to avoid returning NO_SOLUTION_FOUND in most experiments. The exceptions to this pattern were our experiments using the UFRGS GPA dataset to enforce equal opportunity, equalized odds, and predictive equality. In these experiments, the fairness constraints were more strict than in other experiments. Consequently, `Shifty-ttest` tended to return NO_SOLUTION_FOUND. However, in these experiments, all baselines produced models that were unfair above the $\delta = 0.05$ frequency threshold, illustrating the difficulty in satisfying these fairness constraints.

## E.2 UNKNOWN DEMOGRAPHIC SHIFT

Here, we provide additional results evaluating the impact of unknown demographic shift on the fairness properties of trained models. As in the experiments described in Section 3.1.2, we conducted these experiments by uniformly sampling a training dataset from an underlying population (i.e., either the UCI Adult Census dataset or the UFRGS GPA dataset, depending on the experiment), and then evaluate the worst-case impact of unknown demographic shift by considering all possible shifts that agree with the user's input assumptions, as specified by the set $\mathcal{Q}$ (see Section 3). Specifically, the set $\mathcal{Q}$ of possible demographic shifts is given by a set of bounds on the marginal distribution of the demographic attribute after deployment, and the specific bounds used in these experiments are provided in Figure 4. The additional results for experiments using the UCI Adult Census dataset are

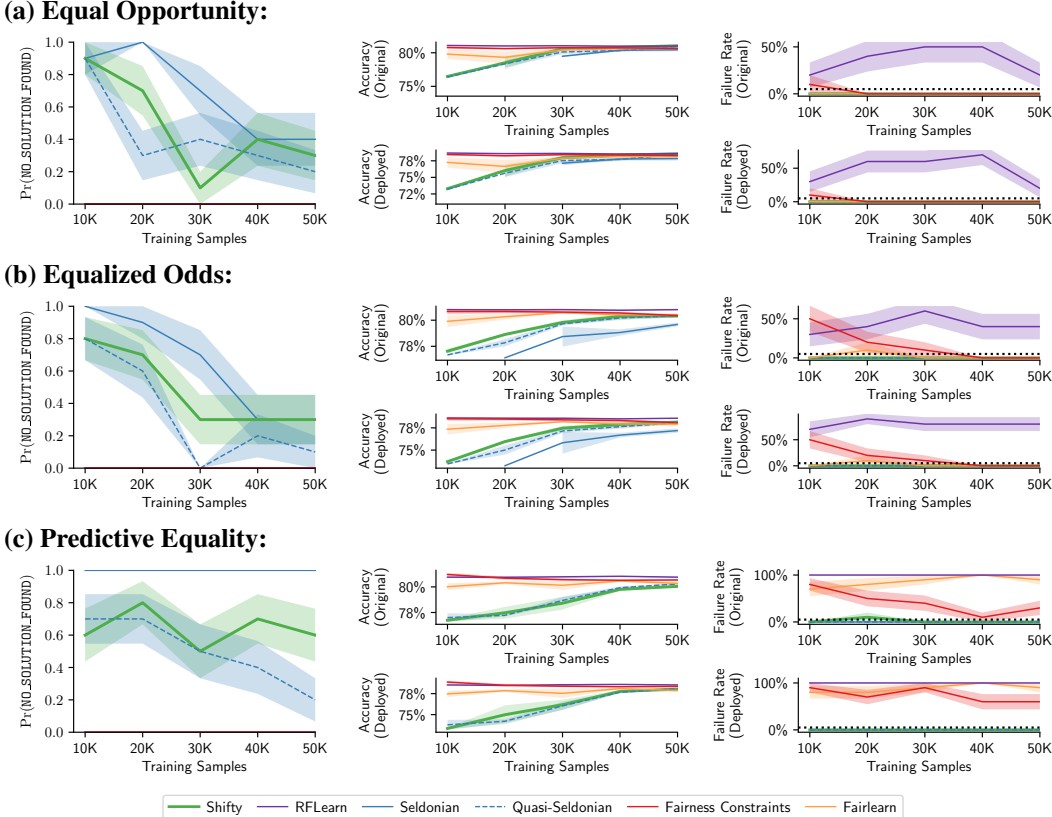

Figure 5: Additional results when enforcing fairness constraints under known demographic shift using the UCI Adult Census dataset.

shown in Figure 7, and full results for the experiments using the UFRGS GPA dataset are shown in Figure 8.

First, we find that the results of these experiments are similar to the results of our experiments using known demographic shift, with the exception that constraints based on unknown demographic shift are more strict than those based on known shift. In all experiments, the output of `Shifty-ttest` was consistently fair, demonstrating that `Shifty` algorithms satisfy Property B even under unknown demographic shift. However, `Shifty-ttest` was more likely to return NO_SOLUTION_FOUND in all experiments than in the experiments with known demographic shift. This is reasonable, since in almost every experiment, we found that models trained using existing baseline algorithms could, in the worst case, be made to behave unfairly depending on the observed demographic shift.

With regards to model accuracy, we found that as long as the fairness constraints were not overly strict, `Shifty-ttest` was able to identify models that were fair and achieved comparable accuracy to baselines, and in some cases, identify models that exceeded the accuracy of baselines by a large margin (see the bottom center plots for the equalized odds experiments in Figure 7). This can be explained by noting that, unlike the baseline algorithms, `Shifty-ttest` selects a candidate model by optimizing an estimate of accuracy after the model is deployed, even if that model performs worse than the baselines on the training data distribution.

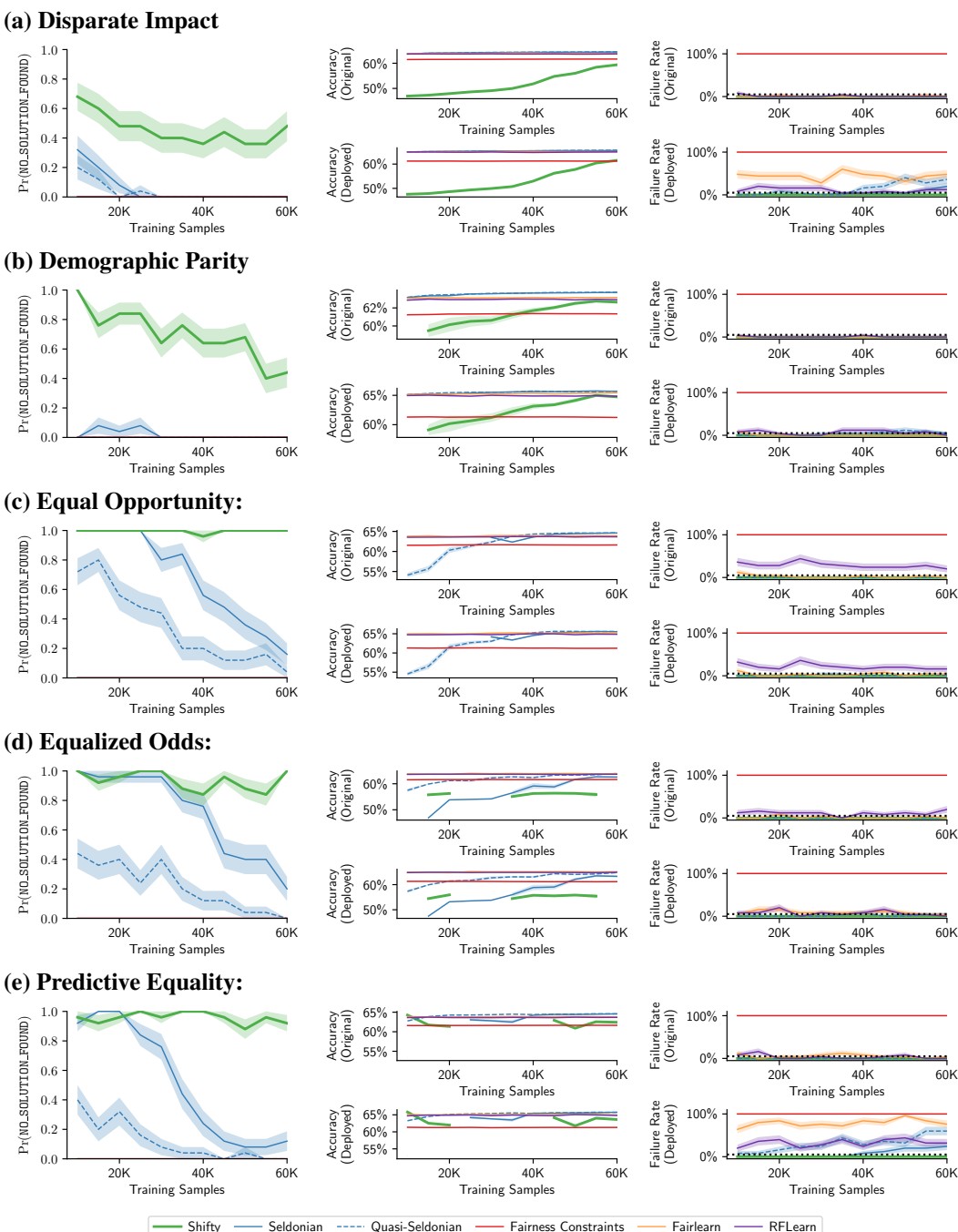

Figure 6: Results when enforcing fairness constraints under known demographic shift using the UFRGS GPA dataset.

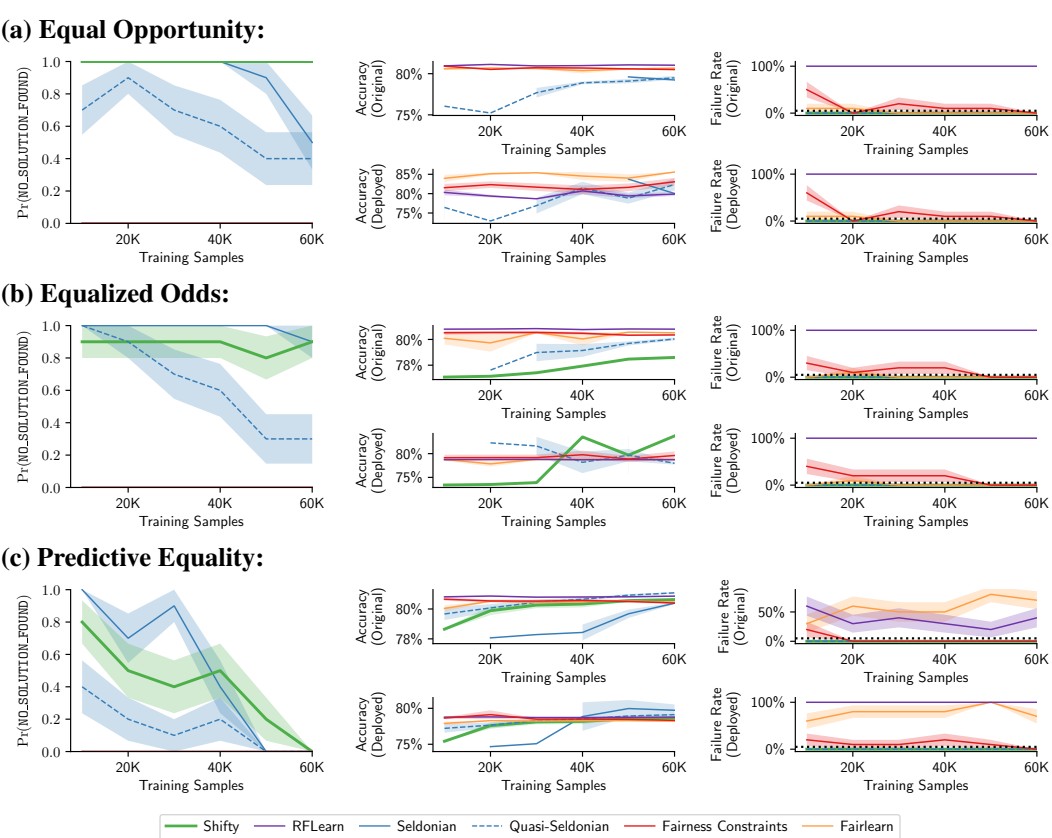

Figure 7: Additional results when enforcing fairness constraints under unknown demographic shift using the UCI Adult Census dataset.

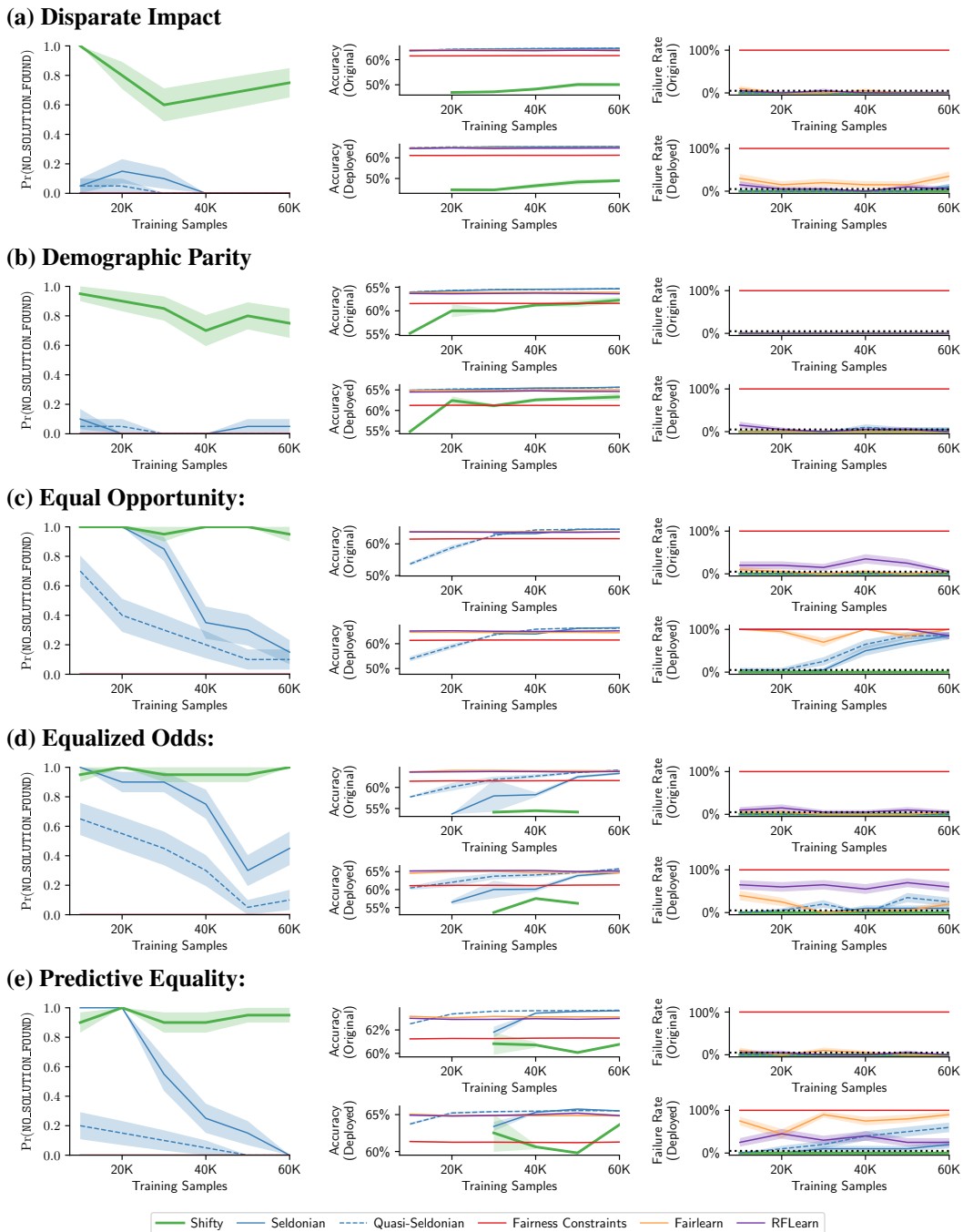

Figure 8: Results when enforcing fairness constraints under unknown demographic shift using the UFRGS GPA dataset.

