# OpenReview forum: "Fairness Guarantees under Demographic Shift"
_ICLR.cc/2022/Conference — ICLR 2022 Poster_

### Official Review · Reviewer_8AcS · 2021-11-02

**Correctness:** 4
**Technical Novelty And Significance:** 3
**Empirical Novelty And Significance:** 3
**Recommendation:** 6
**Confidence:** 3

**Main Review:**

Comments:

-I found the formal definition of demographic shift (page 3) somewhat hard to parse and understand intuitively.

-On top of the previous comment, it is not clear to me why the authors make a distinction between fairness attribute and demographic attribute. What if the college wants to take into account both race and gender as a fairness, legally protected, attribute? Also why is it reasonable to assume that T changes (eq. 2) but the conditional probability (eq. 3) does not? In particular, in what settings it is justified to assume that eq. 3 holds?

-Section 3.1.2 assumes a more realistic scenario. It is hard to think of scenarios where Q would be exactly known so I think this section should be presented as part of the main setup (and not as an extension). However, the description of the method is not very precise. How do the numerical approximation and simplicial homology optimization exactly work in this setting?

-Current appendix A and related work on pages 3-4 could be improved. How does Shifty provide guarantees that the other algorithms cannot? Would it require significant modification of the existing algorithms (and how, if possible) to account for demographic shifts (property B)?

-In terms of writing, the paper is quite clearly written. There are some repetitions in certain places (e.g., section 2)

-In the main contributions list (page 2), I think that the theoretical contribution (#2) is an overstatement. Theorem 1 is rather obvious and no guarantees are provided for the unknown demographic shift case (maybe obtaining distribution-dependent bounds would be possible?)

-Why was only this dataset chosen? Have the authors tried (potentially synthetic) experiments with other datasets (e.g., the law admissions dataset used often in fair ML)?

-How is such a dramatic demographic shift explained in Brazil (fig. 2)?

-The sentence “To evaluate each model’s… E)” about evaluation in Section 4.2 is not very clear.


**Summary Of The Paper:**

The paper focuses on algorithmic fairness in machine learning (ML) and in particular on the problem of demographic shift. The motivation comes from the fact that the distribution of the underlying population might change between training and deployment -- the papers defines this as a shift in the marginal distribution of a single random variable (race, sex, etc.).

The setup considers a classification task, where sensitive information (the “fairness attribute”) might not be legal to use, but is available information. Given a certain event of interest and a target tolerance value \tau (e.g., with respect to sex, the false-positive rate of the model must below \tau% for female), an algorithm is defined to be fair if the algorithm achieves the target value \tau with high probability (given by some \delta parameter). To account for demographic shift, the model further includes a “demographic attribute” T; the distribution of T can change over time, but the conditional distribution must remain the same (by assumption).

Given this setup, the paper introduces Shifty, an ML algorithm that provides high-confidence guarantees that a fairness property will hold even when a demographic shift has occurred. Shifty works as follows. It takes as input a training datasets, some required fairness constraints and a description of demographic shifts. Then, it partitions the dataset to two subsets, finds a model based on the first dataset. Then, it uses Q and the second dataset to build an upper confidence bound for the model after deployment and given demographic shift. It returns only models that are unfair after deployment with probability at most \delta.

The experiments use test scores data from the Brazilian university system and show the satisfactory performance of the proposed algorithm.


**Summary Of The Review:**

I think the paper touches upon an important practical challenge not only in fair ML, but ML models used for decision-making in general. The analysis is natural, straightforward, and technically sound; the most technical aspect is the computation of the upper confidence bound which is standard, so I think there is no particular novelty in its methodological contribution.

That said, I think the conceptual contribution of the paper is still quite new and useful. In the light of the novel application, I thus think the algorithm is quite novel -- nevertheless, it would be helpful if the authors provided a clearer comparison to the most related works and explained how their approach is novel.

Finally, I found the evaluation section convincing and well-executed. I would appreciate additional experiments on -- other widely used in fair ML -- datasets (even synthetic experiments).

---

> ### Author Response · Authors · 2021-11-23
> **Response to Reviewer 4**
>
> Thank you for taking the time to review our work! We hope that our comments below will help clarify some of your concerns!
>
> # Main Comments
>
> > How does Shifty provide guarantees that the other algorithms cannot? Would it require significant modification of the existing algorithms [...] to account for demographic shifts?
>
> The guarantees we refer to in this work are of the form shown in property B, which state that the probability that the training algorithm returns a model that is unfair after demographic shift must be bounded by $\delta$. In practice, the quantities used to define fairness, such as the conditional expected values appearing in the fairness definitions shown in Figure 5, cannot be computed without full knowledge of the deployment distribution. As a result, algorithms can determine that a particular model is empirically fair, even though $g'(\theta) > 0$. To overcome this, Shifty uses a fairness test to ensure that $g'(\theta) \leq 0$ with the required confidence, and returns NO_SOLUTION_FOUND otherwise. Existing approaches to promoting fair outcomes generally do not assess this confidence. Consequently, while these training algorithms may produce models that are empirically fair, they are not proven to satisfy property B. This is especially evident when training on small amounts of training data. In such cases, existing approaches still return a model, despite there not being enough training data to prove that property B holds. One way to modify these algorithms to satisfy property B would be to leverage an independent fairness test, as we propose. Specifically, standard algorithms for promoting fair outcomes could be applied within Shifty to perform candidate selection, and the resulting algorithm would provide fairness guarantees that hold under demographic shift.
>
> > Why is it reasonable to assume that T changes (eq. 2) but the conditional probability (eq. 3) does not?
>
> This is a modelling assumption that the user must consider when applying our approach, similar to the covariate shift assumption made by many existing approaches. The demographic shift assumptions are reasonable when the user is aware of a specific intervention or factor that influences the deployment distribution. For example, a university that would like to use past data from students to predict academic success during admissions the next year might be aware that, due to a recent campaign to promote enrollment among minority communities, it is likely that individuals of certain minority groups will be more likely to apply than they were previously. Here, the university might use Shifty to ensure that their process is fair, despite the shift in demographic.
>
> > Have the authors tried (potentially synthetic) experiments with other datasets (e.g., the law admissions dataset used often in fair ML)?
>
> Thank you for this suggestion! We have conducted new experiments using the UCI Adult dataset, and have updated the paper. These results can be found in Appendix G, and will be incorporated into the main body of the paper.
>
> > It is not clear to me why the authors make a distinction between fairness attribute and demographic attribute.
>
> We distinguish between these attributes to provide generality, but do not assume that they are different. To illustrate why we do not assume that they are the same, consider a demographic parity constraint that states that the positive rate of the classifier must be similar for males and females. Because this constraint is based on expected values that are conditioned on sex, $g(\theta)$ would remain unchanged if the distribution of sex were to shift. However, if a university enforces constraints based on both race and sex, then the prevalence of unfair behavior according to the sex-based constraints may change as the distribution over race changes, and the prevalence of unfair behavior according to the race-based constraints may change as the distribution over sexes changes. Here, the user could set the demographic attribute to be identical to the fairness attribute, namely the combination of sex and race, to ensure that all constraints hold under demographic shift.
>
> > How is such a dramatic demographic shift explained in Brazil (fig. 2)?
>
> The shift shown in Figure 2 was chosen for illustrative purposes, to assess how the fairness properties of various approaches is affected by changes in the deployment environment. In practice, the demographic shift will be specified by the user based on domain expertise. We will modify the paper to make this clear.
>
> > Current appendix A and related work on pages 3-4 could be improved.
>
> We have updated the related work in Appendix A, and will further improve the related work on pages 3-4 to be more clear and comprehensive.
>
> # Other comments/suggestions
>
> Thank you for your feedback on various clarity issues in our paper. We will incorporate these comments and clarify various statements and descriptions throughout our work.

---

> > ### Comment · Reviewer_8AcS · 2021-11-30
> > **thank you for the response**
> >
> > I would like to thank the authors for their response to my review that explains their approach better. I do believe that the paper passes the bar of acceptance. However, I don't think the paper's contribution is too strong to give a score of 8 so I will maintain my current score for now.

---

### Official Review · Reviewer_fKm7 · 2021-11-03

**Correctness:** 3
**Technical Novelty And Significance:** 4
**Empirical Novelty And Significance:** 4
**Recommendation:** 5
**Confidence:** 4

**Main Review:**

The paper tackles an interesting and challenging problem. The idea of separating the tasks in a parallel sense is interesting as they divide the problem into the candidate selection and searching for the high-confidence upper bound. The main issue, however, is the data distribution part that it seems that they didn’t describe its procedure, hence, I am not sure whether it was a fair sample selection in which both of the Df and Dc are representative of the source data.

In the introduction, the authors claimed that they allow users to select fairness notion(s) from a large class depending on the application domain. However, according to their approach, it is not clear how they can account for individual fairness. Can the authors elaborate on this?

Furthermore, the authors acknowledge in section D of supplementary material, there is an assumption that each parameter’s intervals are independent, which is often false. Can the authors explain such justifications.

Their method of identifying the distribution shift in both known and unknown shifts relies on user-provided information. Indeed, it is an assumption to the problem which may hold untrue.  Would it be possible to detect distribution shift at the deployment and change the model accordingly? Why there is a need to train models for various unknown distribution shifts? ( It is obvious that considering various shifts has a huge impact on the performance (according to their results in Section 4.2)?

The structure of the evaluation section is not convincing to me, and the comparison with in-processing fairness approaches does not seem fair. The only valid comparison is wrt to the fairness guarantees. However, it is not clear to me why the authors compared their approach under distribution shift with other methods that do not claim to be fair when the distributions of the train and test sets are different. Also, given the results presented for the experiment with unknown distribution shift,  it seems the Seldonian algorithm performs better that their approach. I'd suggest the authors to compare their approach with methods proposed for concept shift, also it will be useful to add at-least one more dataset for the evaluation to make sure that these results are consistent in various contexts.

**Summary Of The Paper:**

The authors propose an approach called SHIFTY that provides high-confidence fairness guarantees when the distribution of training and deployment is different. They proposed two approaches to tackle the problem, one based on known distribution shift and another one for when the distribution shift is unknown. They evaluated their approach on a dataset for student success prediction and compared their approach with SOTA in-processing fairness approaches.

**Summary Of The Review:**

I have found the main idea of the paper interesting, however, I have some doubts about their proposed method and the way that they evaluated their approach and compared it with other methods (see previous section).
Also, I suggest the authors to focus only on group fairness as it is not clear how their approach can be used for individual fairness methods.

---

> ### Author Response · Authors · 2021-11-23
> **Response to Reviewer 3**
>
> We appreciate your feedback! We hope our comments below will clarify some of your questions.
>
> > The data distribution part that it seems that they didn’t describe its procedure, hence, I am not sure whether [...] Df and Dc are representative of the source data.
>
> Data partitioning splits the training set into $D_f$ and $D_c$ randomly, without replacement. We shuffle the input data, assign the first half to $D_c$, and the other half $D_f$. Since the input data is assumed to be drawn i.i.d. from the training distribution, this ensures that $D_f$ and $D_c$ also consist of i.i.d. samples from the training distribution. We will clarify this in the paper.
>
> > I'd suggest the authors to compare their approach with methods proposed for concept shift, [and also add] one more dataset for the evaluation
>
> We have updated our paper to include results using the UCI Adult dataset, and evaluated a baseline that promotes fair outcomes under covariate shift. While there are several approaches that promote fair outcomes under distribution shift, most can only be applied for specific types of shift that differ from demographic shift or require access to data from the deployment distribution. We have updated our paper to include a description of these approaches in Appendix A.
>
> > There is an assumption that each parameter’s intervals are independent, which is often false.
>
> Violation of this assumption does not impact the validity of the upper bound on $g'(\theta)$ or Shifty's guarantees. When this constraint is violated, the upper bound on $g'(\theta)$ becomes larger than necessary, but this does not result in unfair models being erroneously returned.
>
> If the dependence between parameters is known, one can construct tight, high-confidence upper bounds on $g'(\theta)$, resulting in an algorithm that satisfies property B. However, if the model of dependence is incorrect, the algorithm will not provably satisfy property B. Unfortunately, the dependence between random variables defining fairness can usually only be modeled approximately. Rather than assume that these dependencies are known, we compute upper bounds as though each parameter's confidence intervals are independent. The resulting bounds are larger but still valid, and ensure that Shifty's guarantees do not depend on perfect knowledge of the dependence between the random variables used to define fairness. We will update the paper to clarify this point.
>
> > It is not clear how they can account for individual fairness. Can the authors elaborate on this?
>
> Shifty can be implemented to enforce individual fairness constraints. While Shifty-ttest employs confidence intervals that are appropriate for producing bounds on expected values, individual fairness definitions are often based on the maximum of a functional over pairs of similar individuals. Using strategies for estimating these maximums, one can implement a Shifty algorithm that enforces individual fairness. We will update our paper to discuss this point.
>
> > Their method of identifying the distribution shift [...] relies on user-provided information. Would it be possible to detect distribution shift at the deployment and change the model?
>
> We assume that no data is available from the deployment distribution, so it is not possible to detect distribution shift. For example, if a university would like to use data from the current school year to make fair predictions during admissions for the following year, samples from the deployment distribution are unavailable. If deployment data is available, then this data can be used to estimate the future demographic distribution, and Shifty can be applied. We will discuss this in the paper.
>
> > Why there is a need to train models for various unknown distribution shifts?
>
> Shifty trains a model that is shown to be fair on all demographic distributions in $\mathcal{Q}$ with high confidence. Therefore, Shifty provides guarantees that hold for any demographic distribution, provided it is in $\mathcal{Q}$. This is useful if the user has domain knowledge that can describe which demographic distributions are plausible. It is also useful when the demographic distribution might change over time. Consider a university that trains a classifier to predict student success, which will be used for several years. Shifty's guarantees will continue to hold over time, provided the demographic distribution each year is within $\mathcal{Q}$.
>
> > [Under unknown demographic shift] it seems the Seldonian algorithm performs better.
>
> Seldonian algorithms achieved higher accuracy than Shifty in our Brazil experiments, but frequently violated the fairness constraints, unlike Shifty. Therefore, Seldonian algorithms cannot be used when unfair outcomes must be avoided. Also, this difference in accuracy is dependent on the dataset. In our updated experiments using the UCI Adult dataset (Appendix G), Shifty matched or outperformed the accuracy of the Seldonian algorithms while maintaining its fairness guarantees.

---

> > ### Comment · Reviewer_fKm7 · 2021-11-30
> > **Thank you**
> >
> > I thank the authors for their response to my review.
> > Adding new results and running experiments on a new dataset, i.e, Adult, with a comparison with existing methods on covariate shifts would definitely improve the paper.
> > Also, it is important that the authors add the discussion on how their approach can extend to individual fairness notion in the final version of the paper as promised.
> > Regarding the unknown distributions, I am still not convinced by their example. As for the case of predicting students' success, the cost of re-training the model on new data distribution (or even using meta-learning and fine-tuning) is less than the cost of having a fair but non-accurate model. I can imagine that the fairness guarantee is important under various distribution shifts only for special cases that the cost of training on new distribution is high/impossible. I'd suggest discussing/running experiments to compare their approach with meta-learning approaches with fairness guarantees.

---

### Official Review · Reviewer_D4Vj · 2021-11-03

**Correctness:** 3
**Technical Novelty And Significance:** 3
**Empirical Novelty And Significance:** 3
**Recommendation:** 6
**Confidence:** 4

**Main Review:**

Positives:

1. The paper is generally well-written and easy to follow.
2. The problem studied is definitely interesting to the community and one of the main challenges with the deployment of models in practical settings.
3. The high-confidence upper bounds on unfairness in the deployment setting present an interesting approach to ensuring fairness guarantees in the deployment setting.

Negatives:
1. The authors are missing some parallel works with ensuring fairness guarantees with general distribution shift [1,2,3].
2. Certain design choices are missing clear explanations such as how was the interpolation factor of 0.3 decided?

Additional concerns:
1. Can the assumptions about $g(\theta)$ be extended to settings where there needs a comparison with respect to another demographic group rather than ensuring it to be under some threshold?
2. It is not exactly clear how the approach can be extended to definitions such as individual fairness.
3. In order to better assess the quality of the fair predictions in the deployment setting, it would be helpful to have some empirical analysis when multiple definitions of fairness that may not be compatible are suggested by the user.

References:
1. Singh, H., Singh, R., Mhasawade, V., & Chunara, R. (2021, March). Fairness violations and mitigation under covariate shift. In Proceedings of the 2021 ACM Conference on Fairness, Accountability, and Transparency (pp. 3-13).
2. Du, W., & Wu, X. (2021). Robust Fairness-aware Learning Under Sample Selection Bias. arXiv preprint arXiv:2105.11570.
3. Dai, J., & Brown, S. M. Label Bias, Label Shift: Fair Machine Learning with Unreliable Labels.

**Summary Of The Paper:**

The paper presents a class of algorithms for ensuring fairness guarantees when the deployment data is susceptible to a demographic shift, termed as a marginal shift in demographic attributes such as gender or race, in comparison to the training data. Results are provided for two settings, 1) when the exact demographic shift in the deployment setting is known and 2) when the demographic shift in the deployment environment is unknown. This is done by computing a high-confidence upper bound on the prevalence of unfair behavior in the deployment environment, presently done using the Student's t-Test. Comparisons are made with Seldonian and Quasi-Seldonian algorithms, Fairlearn, and Fairness Constraints. The empirical analysis is performed on the university dataset to predict GPA using scores, with gender as the fairness attribute and student's race as the demographic attribute, the marginal distribution of which changes across the training and deployment settings.

**Summary Of The Review:**

The contributions of the paper are clear. As the paper is missing some related works, the baseline comparisons can be further extended to assess other factors as suggested.

---

> ### Author Response · Authors · 2021-11-23
> **Response to Reviewer 2**
>
> Thank you for your comments and suggestions! We appreciate your feedback, and hope our comments below will answer some of the questions and concerns you raised.
>
> > The authors are missing some parallel works with ensuring fairness guarantees with general distribution shift.
>
> We have updated the related work section in Appendix A to discuss these approaches, and will update the discussion of related work in the main body of the paper as well. In addition, we reviewed these approaches to determine whether or not we could compare to them in our evaluations. One of these, RFLearn [2], is suitable for a direct comparison in our evaluation. [3] assumes a substantially different form of distribution shift (label shift and bias) from demographic shift, and would not work with our shift, and [1] requires access to a causal graph of the problem, which is a significant limitation of the applicability of that approach. In particular, no causal graph exists in our evaluation setting. As a result, we cannot directly evaluate against [1] and [3]. RFLearn [2] promotes fair outcomes under covariate shift; although covariate shift and demographic shift are not identical (see, for example, our updated related work section in Appendix A), they are similar in certain settings. Unlike our approach, RFLearn does not provide guarantees similar to those provided by Shifty. To evaluate this, we conducted a new set of experiments using the UCI Adult dataset used in [2] to evaluate Shifty and multiple baseline approaches, including RFLearn. Full results can be found in Appendix G of our updated draft; our findings indicate that RFLearn tends to frequently violate fairness constraints, whereas Shifty does not.
>
> > Certain design choices are missing clear explanations such as how was the interpolation factor of 0.3 decided?
>
> We will update our draft to make the choice of interpolation factor and other design choices more clear. Regarding the interpolation factor, we chose 0.3 to simulate a possible set of demographic distributions that a user might want to be fair with respect to. Crucially, the choice of interpolation factor defines the problem setting used in our evaluation but does not impact the valilidty of our results. In practice, the set $\mathcal{Q}$ would be specified by the user based on domain knowledge, and would not be constructed using an interpolation scheme. We will modify the paper to clarify that this strategy for defining $\mathcal{Q}$ is for illustrative purposes.
>
> > Can the assumptions about $g(\theta)$ be extended to settings where there needs a comparison with respect to another demographic group rather than ensuring it to be under some threshold?
>
> This is certainly possible, and several of the fairness definitions we consider in our experiments are constructed in this way. For example, our constraints based on demographic parity enforce the positive rate of the classifier to be similar for different races.
>
> > It is not exactly clear how the approach can be extended to definitions such as individual fairness.
>
> Individual fairness constraints can be enforced using Shifty algorithms, but pose unique challenges. Specifically, they require estimating the maximum of certain functionals, rather than expected values. As a consequence, the approach taken when deriving Shifty-ttest, which computes bounds on expected values, cannot be used. However, given appropriate strategies for estimating these maximums, a Shifty algorithm can be designed that would provide the high-confidence guarantees presented. We will update our paper to describe how Shifty can be used to enforce these definitions.
>
> > In order to better assess the quality of the fair predictions in the deployment setting, it would be helpful to have some empirical analysis when multiple definitions of fairness that may not be compatible are suggested by the user.
>
> If multiple conflicting constraints are specified by the user, Shifty will consistently return NO_SOLUTION_FOUND. On the other hand, existing baselines will still output a model, but this model will violate one or more of the fairness constraints. We consider this to be desirable behavior. For example, if deploying a model that fails to meet the fairness constraints would result in negative repercussions for stakeholders, it is favorable for the training algorithm to signal to the user that no model is able to satisfy the fairness constraints, rather than return a model that is eventually deployed and results in undesirable or dangerous outcomes. We will conduct experiments to empirically verify this claim.

---

> > ### Comment · Reviewer_D4Vj · 2021-11-29
> > **Thank you for the response**
> >
> > I thank the authors for the clarifications. I have carefully read the authors' responses as well as other comments. The authors have addressed most of my concerns, I keep my score but believe that the problem is interesting and the work paves an important direction.

---

### Official Review · Reviewer_3Aub · 2021-11-04

**Correctness:** 4
**Technical Novelty And Significance:** 3
**Empirical Novelty And Significance:** 3
**Recommendation:** 8
**Confidence:** 5

**Main Review:**

The work provides a conceptually clear and flexible framework for learning fair classifiers under demographic shifts. The organization of the concepts (Seldonian framework, problem description, fairness tests) is great and the writing is clear. Isolating the problem to performing the fairness test under an unknown distribution helps to understand the proposed method. The flexibility of the framework to accommodate different fairness measures is an advantage. The probabilistic guarantee for fairness violation is a good feature absent from many fair learning approaches for this particular problem.
My main issues are (1) with the setup of controlling only fairness and not the accuracy, and (2) the seemingly large number of samples required for good accuracy for the method. Some related work is also missing which can be easily addressed. Few details on the scope of the demographic variable (discrete vs continuous) and relation to covariate shifts need clarity. I mention these issues in detail below.

---

## Questions to address in the response:

1. Why is the objective of controlling test unfairness while controlling train error justified, as opposed to attending to both quantities for test set? As pointed out, determination of the fairness test is impacted by demographic shift. So, is the classification loss at deployment time that is to be computed for the shifted distribution. Does this accounted somehow in the algorithm? One strategy is similar to the proposed fairness test procedure to note that loss conditional on T remains the same and somehow importance weighting or bounding the loss. I am wondering if I am missing this important detail in the algorithm on how training and deployment error are related to each other.

2. Does the demographic shift assumption rely on demographic attribute space \mathcal{T} to be discrete? The specification of marginal shift in terms of lower-upper bounds on each probability of each value of T suggests that. Similarly, conditioning set \zeta around Eq (4) is assumed to be discrete, thus, excluding some fairness metrics, which is fine, but the allowed domains for variables should be specified. Please mention if discreteness is a limiting assumption for the proposed algorithm.

3. Accuracy of Shifty seems to be severely impacted even with known shifts when there are <100K data points in middle plots of Figure 3. Under known shift, the test fairness constraint is not harsher than the fairness constraint on the train set. What is the reason for the drop? Also, it is more practical to have much fewer data points such as of the order of 10K. At least, the sizes of fairness datasets used in literature are in the same or lower range. Does the severe drop in accuracy observed in only this particular dataset of exam scores or others too? It would be worthwhile to either include results for lower sample sizes on this and preferably other datasets, or empirically study and discuss issues limiting the sample efficiency for future work to address.

---

## Minor questions that do not affect my review:
4. In Algorithm 1, does the candidate model \thetha_c in step 2 appear in inputs to fairness testing in step 3?

5. I missed the description of how the candidate models are enumerated and tested one by one. Can you please clarify what is the output of the candidate model selection step 2 in Algorithm 1.

---

## Related work

There have been multiple studies on fairness guarantees under shifts which have been omitted. They are making somewhat different, in some cases weaker, assumptions and have different guarantees. It would be good to discuss differences from these works.

- Biswas and Mukherjee 21 https://dl.acm.org/doi/abs/10.1145/3461702.3462596
- Rezaei et al. 21 https://arxiv.org/abs/2010.05166
- Schumann et al. 19 https://arxiv.org/abs/1906.09688
- Coston et al. 19 https://dl.acm.org/doi/abs/10.1145/3306618.3314236
- Singh et al. 21 https://arxiv.org/abs/1911.00677
- Dai and Brown 20 https://dynamicdecisions.github.io/assets/pdfs/29.pdf

If any of these works have the same problem setup such as Rezaei et al. 21, please consider comparing against them.

---

## Suggestions

Please mention if the demographic shift assumption is a type of covariate shift, which is a better known term in literature or how does it differ.

Please describe how the result in Theorem 2 is important and used to support the method.

Please add the reason for choosing the particular numerical optimizer from Endres et al. 2018. Is U_ttest a non-smooth function?

A possible dataset to experiment is the new Adult dataset derived from US Census which has distribution shifts, perhaps demographic and other shifts. See Folktables package https://github.com/zykls/folktables.

**Summary Of The Paper:**

The paper proposes a method for controlling unfairness of a model when the test distribution may differ in the marginal distribution of a feature such as gender, race. It derives a statistical test for checking unfairness on the unknown test distribution via importance weighting combined with user-input on extent of the shift. Experiments on an academic performance dataset shows that the approach achieves low failure rate of learning an unfair classifier.

**Summary Of The Review:**

The work presents a novel and clear approach to the problem of fair learning under shifts. There are some open issues on the problem setup and experimental results, which I would like authors to respond to. In total, the work is a technically strong contribution to the nascent literature on the problem.

---
From the newly made changes and clarifications in the response, my concerns are addressed. I highly encourage authors to move the results on new dataset, discussion on related work and continuous demographic attributes to the main text.

---

> ### Author Response · Authors · 2021-11-23
> **Response to Reviewer 1**
>
> Thank you for your detailed feedback!
>
> # Comments on main questions
>
> > Why is the objective of controlling test unfairness while controlling train error justified, as opposed to attending to both quantities for test set?
>
> As you mentioned, optimizing for accuracy on the deployment distribution when the demographic shift is known can be done in the same way that we estimate the prevalence of unsafe behavior after deployment, and is straightforward to implement in our codebase. We did not originally consider this because the main objective of our work was to propose strategies for establishing high-confidence fairness guarantees that hold under demographic shift. Nonetheless, in response to your comment, we are currently running experiments that optimize for deployment accuracy directly, although due to time constraints, these results are not yet available.
>
> > Does the demographic shift assumption rely on demographic attribute space $\mathcal{T}$ to be discrete? [...] Similarly, conditioning set $\zeta$ around Eq (4) is assumed to be discrete, thus, excluding some fairness metrics, which is fine, but the allowed domains for variables should be specified.
>
> The demographic shift assumption does not rely on $\mathcal{T}$ or $\mathcal{S}$ being discrete sets. First, $S$ can be continuous as long as $\zeta$ corresponds to a Boolean event to ensure that any conditional probabilities, $\Pr(\cdot|\zeta)$ are well-defined). For example, $\zeta = (S < 0.5)$ and $\zeta = (0 \leq S < 0.5)$ are valid conditions that might be used if $S$ is continuous. Next, $T$ can also be continuous, although this complicates the process of specifying $\mathcal{Q}$ and, in the case of unknown demographic shift, optimizing $\hat{U}_{\text{ttest}}$ over $q \in \mathcal{Q}$.
>
> > Accuracy of Shifty seems to be severely impacted even with known shifts when there are <100K data points in middle plots of Figure 3. [...] What is the reason for the drop? Also, it is more practical to have much fewer data points such as of the order of 10K. [...] Does the severe drop in accuracy observed in only this particular dataset of exam scores or others too?
>
> Great question! The accuracy of Shifty generally degrades compared to alternative approaches for small training set sizes because Shifty must establish confidence that a model will behave fairly in order for it to be returned. Under demographic shift, this confidence is determined using importance-weighted samples, which have higher variance than those obtained directly from the training or deployment distributions. Consequently, for problems in which fairness is at odds with improved accuracy, Shifty tends to return low-accuracy models that are easier to verify using small amounts of data. For this reason, when training data is scarce, the accuracy of Shifty-trained models tends to be lower than that of models trained using Seldonian algorithms (which do not provide guarantees that hold under demographic shift and therefore base the fairness test on lower-variance samples from the training set) or baselines (which do not need to compute confidence because they do not provide high-confidence fairness guarantees). However, we have recently conducted experiments using the UCI Adult dataset (see Appendix G for results), and found that there, the accuracy drop associated with using Shifty was very small. We speculate that, depending on the application, high-confidence fairness guarantees may not be possible to obtain without some loss in accuracy, and plan to investigate this phenomenon in future work.
>
> To better assess the size of this accuracy drop, and to test if it occurs on other datasets, we have conducted preliminary experiments using the UCI Adult dataset, which we have included in Appendix G in our updated draft. In addition, we modified our experimental design to specifically evaluate training set sizes between 10k-50k.
>
> # Related Work
>
> Thanks for your suggestions on these related work! Based on your suggestions, we have updated the related work section of our paper (Appendix A) to include these references, with explanations of how they relate to our approach. We will also update the related work in the main body of the paper to discuss these references.
>
> # Comments on minor questions/suggestions
>
> Thank you for your additional comments and suggestions! We will follow this response with another that answers and clarifies these concerns, and incorporate these suggestions into our next draft.

---

### Decision · Program_Chairs · 2022-01-20

**Decision:**

Accept (Poster)

**Comment:**

The paper studies the setting of group-based fairness under the so-called demographic shift, where the marginal distribution of the data remains the same conditional on the subgroup but the subgroup distribution can change. It provides a class of algorithms which give high confidence guarantees under demographic shift in both the known and unknown shift setting.

Overall the paper is a worthwhile contribution: it provides a new angle to the important problem of group-based fairness with good theoretical and empirical results.